# Cavβ1 regulates T cell expansion and apoptosis independently of voltage-gated Ca$^{2+}$ channel function

Serap Erdogmus[1,11], Axel R. Concepcion[1,11], Megumi Yamashita[2], Ikjot Sidhu[1], Anthony Y. Tao[1], Wenyi Li [1], Pedro P. Rocha [3,4], Bonnie Huang [5,6], Ralph Garippa[7], Boram Lee[8], Amy Lee[9], Johannes W. Hell [8], Richard S. Lewis[10], Murali Prakriya[2 ✉] & Stefan Feske [1 ✉]

TCR stimulation triggers Ca$^{2+}$ signals that are critical for T cell function and immunity. Several pore-forming α and auxiliary β subunits of voltage-gated Ca$^{2+}$ channels (VGCC) were reported in T cells, but their mechanism of activation remains elusive and their contribution to Ca$^{2+}$ signaling in T cells is controversial. We here identify Ca$_V$β1, encoded by *Cacnb1*, as a regulator of T cell function. *Cacnb1* deletion enhances apoptosis and impairs the clonal expansion of T cells after lymphocytic choriomeningitis virus (LCMV) infection. By contrast, *Cacnb1* is dispensable for T cell proliferation, cytokine production and Ca$^{2+}$ signaling. Using patch clamp electrophysiology and Ca$^{2+}$ recordings, we are unable to detect voltage-gated Ca$^{2+}$ currents or Ca$^{2+}$ influx in human and mouse T cells upon depolarization with or without prior TCR stimulation. mRNAs of several VGCC α1 subunits are detectable in human (Ca$_V$3.3, Ca$_V$3.2) and mouse (Ca$_V$2.1) T cells, but they lack transcription of many 5' exons, likely resulting in N-terminally truncated and non-functional proteins. Our findings demonstrate that although Ca$_V$β1 regulates T cell function, these effects are independent of VGCC channel activity.

[1] Department of Pathology, NYU Grossman School of Medicine, New York, NY, USA. [2] Department of Pharmacology, Northwestern University, Chicago, IL, USA. [3] Unit on Genome Structure and Regulation, National Institute of Child Health and Human Development, National Institutes of Health, Bethesda, MD, USA. [4] National Cancer Institute, NIH, Bethesda, MD, USA. [5] National Institute of Allergy and Infectious Disease, Bethesda, MD, USA. [6] National Human Genome Research Institute, Bethesda, MD, USA. [7] Department of Cancer Biology & Genetics, Memorial Sloan Kettering Cancer Center, New York, NY, USA. [8] Department of Pharmacology, University of California, Davis, CA, USA. [9] Department of Neuroscience, University of Texas-Austin, Austin, TX, USA. [10] Department of Molecular and Cellular Physiology, Stanford University, Stanford, CA, USA. [11] These authors contributed equally: Serap Erdogmus, Axel R. Concepcion. ✉email: m-prakriya@northwestern.edu; feskes01@nyumc.org

Changes in intracellular $Ca^{2+}$ concentration ($[Ca^{2+}]_i$) are essential for signal transduction in all eukaryotic cells including T lymphocytes[1,2]. The best characterized $Ca^{2+}$ influx pathway in T cells is store-operated $Ca^{2+}$ entry (SOCE) mediated by $Ca^{2+}$ release-activated $Ca^{2+}$ (CRAC) channels encoded by ORAI1 and its homologue ORAI2. TCR stimulation results in the generation of the second messenger inositol 1,4,5-trisphosphate ($IP_3$), the opening of $IP_3$ receptor channels and $Ca^{2+}$ release from the endoplasmic reticulum (ER). $Ca^{2+}$ efflux from the ER causes the activation of STIM1 resulting in its binding to ORAI1 and opening of CRAC channels. CRAC channels are critical for T cell function and immunity to infection as evidenced by the immunodeficiency of CRAC-deficient human patients and mice[2].

Other $Ca^{2+}$ channels that have been proposed to mediate $Ca^{2+}$ influx in T cells are voltage-gated $Ca^{2+}$ channels (VGCCs). They are critical for $Ca^{2+}$ signaling in excitable cells such as neurons, cardiomyocytes, skeletal muscle and secretory cells[3], but their function in T cells is less well established. VGCCs are divided into three groups: high-voltage activated (L-Type: Cav1.1, Cav1.2, Cav1.3, Cav1.4; N-Type: Cav2.2; P/Q-Type: Cav2.1), intermediate-voltage activated (R-Type: Cav2.3) and low-voltage activated (T-Type: Cav3.1, Cav3.2, Cav3.3) channels[3,4]. VGCCs are composed of a $Ca^{2+}$ conducting, pore-forming α1 subunit and several auxiliary β, $α_2δ$- and γ-subunits. The $α_1$ subunits are composed of four domains (I–IV), each consisting of 6 α-helical transmembrane domains (S1–S6). S1–S4 form the voltage sensing domain (VSD) with S4 containing positively charged amino acids that sense changes in membrane potential, while the S5–S6 subunits constitute the ion conduction pore and selectivity filter[3,5]. Each VGCC has one β-subunit, which binds via its $α_1$-binding pocket (ABP) to the cytosolic $α_1$−interacting domain (AID) in the linker region between domains I and II of the $α_1$ subunit. The four β subunit homologues (β1–β4) increase the plasma membrane expression of $α_1$ subunits, enhance $Ca^{2+}$ currents and modulate the voltage-dependence and kinetics of activation and inactivation of VGCCs[6,7]. The α1 subunit furthermore binds to the extracellular $α_2δ$-subunit and the transmembrane γ-subunit consisting of four transmembrane domains. Mutations in VGCCs and altered $Ca^{2+}$ influx are associated with a plethora of human diseases including cardiac arrhythmias and psychiatric diseases (Cav1.2), autism spectrum disorder and primary aldosteronism (Cav1.3), various X-linked retinal disorders (Cav1.4), familial hemiplegic migraine (Cav2.1), epilepsy (Cav1.3, Cav2.1, Cav3.2) and several forms of ataxia (Cav2.1, Cav3.1)[8–11]. Similarly, deletion or mutation of $α_1$ and auxiliary VGCC subunits in mice has been reported to cause a large spectrum of neurological, cardiovascular, musculoskeletal and endocrinological phenotypes[7,8,12,13]. $Ca^{2+}$ channel blockers targeting VGCCs such as nimodipine, verapamil and diltiazem are in wide clinical use for the treatment of arterial hypertension[5]. A common denominator of human diseases associated with VGCC mutations, phenotypes of genetically altered mice and the effects of $Ca^{2+}$ channel blockers is that they originate from the altered function of excitable cell types such as neurons, cardiomyocytes or secretory cells including pancreatic β cells or adrenal chromaffin cells[14].

Several studies have reported that VGCCs are modulating immune responses and $Ca^{2+}$ signaling in T cells by using $Ca^{2+}$ channel blockers, RNA interference (RNAi) and genetic deletion of various $α_1$ and β subunits in mice[1]. RNAi-mediated deletion of Cav1.2 and Cav1.3, or β subunits with antisense oligonucleotides, showed reduced TCR-induced $Ca^{2+}$ influx, cytokine production and experimental asthma in $CD4^+$ T cells polarized into T helper 2 (Th2) cells[15,16]. T cells from $Cacna1f^{-/-}$ mice lacking Cav1.4, which is highly expressed in the retina, had reduced $Ca^{2+}$ influx

and $Ba^{2+}$ currents in T cells and showed a defect in the function, development and survival of naïve T cells and in T cell responses to intracellular pathogens in vivo[17,18]. Genetic deletion of the T-type VGCC Cav3.1 in mice had no effect on TCR-induced $Ca^{2+}$ influx in T cells despite reduced low-voltage activated $Ca^{2+}$ currents[19]. However, Cav3.1-deficient mice were protected from experimental autoimmune encephalomyelitis (EAE), which was associated with reduced numbers of IFN-γ and GM-CSF producing T cells in vivo and defects in Th17 cell function in vitro including $Ca^{2+}$ influx, NFAT activation, and the expression of RORγt and IL-17A[19]. In addition, several studies have implicated β subunits of VGCCs in T cell function. For example, T cell-specific deletion of Cavβ2 resulted in a severe defect in T cell development due to impaired thymocyte proliferation and survival[20]. T cells from $Cacnb2^{-/-}$ (Cavβ2 knockout) mice had moderately reduced TCR-induced $Ca^{2+}$ influx, which was associated with loss of Cav1.2 and Cav1.3 protein expression[20]. Similarly, T cells of lethargic mice with a spontaneously occurring mutation in Cacnb4 (encoding Cavβ4) exhibited splenic and thymic involution and lymphocytopenia[21,22]. TCR-induced $Ca^{2+}$ influx was moderately reduced in T cells of Cavβ4 mutant mice and those from mice with targeted deletion of Cavβ3[23]. Although T cells of $Cacnb3^{-/-}$ mice had moderately reduced TCR-induced $Ca^{2+}$ influx, the survival of naive $CD8^+$ T cells was profoundly impaired due to altered expression of pro- and antiapoptotic genes[24]. Cavβ3 deficiency was associated with loss of Cav1.4 protein expression, suggesting that Cavβ3 may regulate $Ca^{2+}$ influx in T cells through Cav1.4[24]. Collectively, these studies suggest that VGCCs contribute to T cell development and function, potentially by regulating TCR-stimulated $Ca^{2+}$ signaling. VGCC as $Ca^{2+}$ channels in T cells, however, are not universally accepted[25], and biophysical evidence of VGCC currents in T cells is limited[17,19]. Given the non-excitable nature of T cells it also remains unclear how VGCCs are activated in the context of T cell activation.

In this study, we identify Cacnb1 (Cavβ1) as a regulator of T cell function. Although Cavβ1 is well-studied in skeletal muscle, where it modulates excitation/contraction coupling[26], its function in T cells has not been reported. Using a pooled shRNA screen to identify ion channels that regulate T cell responses to viral infection in vivo, we find that deletion of Cacnb1 impairs the clonal expansion of antigen specific T cells after viral infection in vivo by enhancing T cell apoptosis. Cacnb1 deletion does not affect TCR-induced $Ca^{2+}$ signaling and production of $Ca^{2+}$ regulated cytokines, suggesting that its function in T cells differs from its canonical one in excitable cells modulating the function of VGCCs. Indeed, a detailed search for voltage-gated $Ca^{2+}$ currents and $Ca^{2+}$ signals in human and mouse T cells fails to provide evidence for the existence of functional VGCCs in T cells. While mRNAs of several VGCC α1 subunits are detectable in T cells by RNA-Seq (Cav3.3, Cav3.2 and Cav2.1), these transcripts are incomplete, and lack expression of multiple 5′ exons that encode the first two (of four) $Ca_V$ domains. We conclude that full-length transcripts of $α_1$ subunits of VGCCs are not expressed in T cells, providing an explanation for the absence of VGCC currents and $Ca^{2+}$ influx upon depolarization in T cells.

## Results

**shRNA screen in vivo identifies *Cacnb1* as a VGCC subunit required for clonal expansion of T cells during LCMV infection.** To identify ion channels and transporters (ICTs) that regulate T cell function and T cell-mediated immunity during viral infection in vivo, we generated a library of 658 ICTs and regulatory factors, of which 602 ICTs were annotated in both mouse and human genomes. These ICTs were analyzed for their mRNA

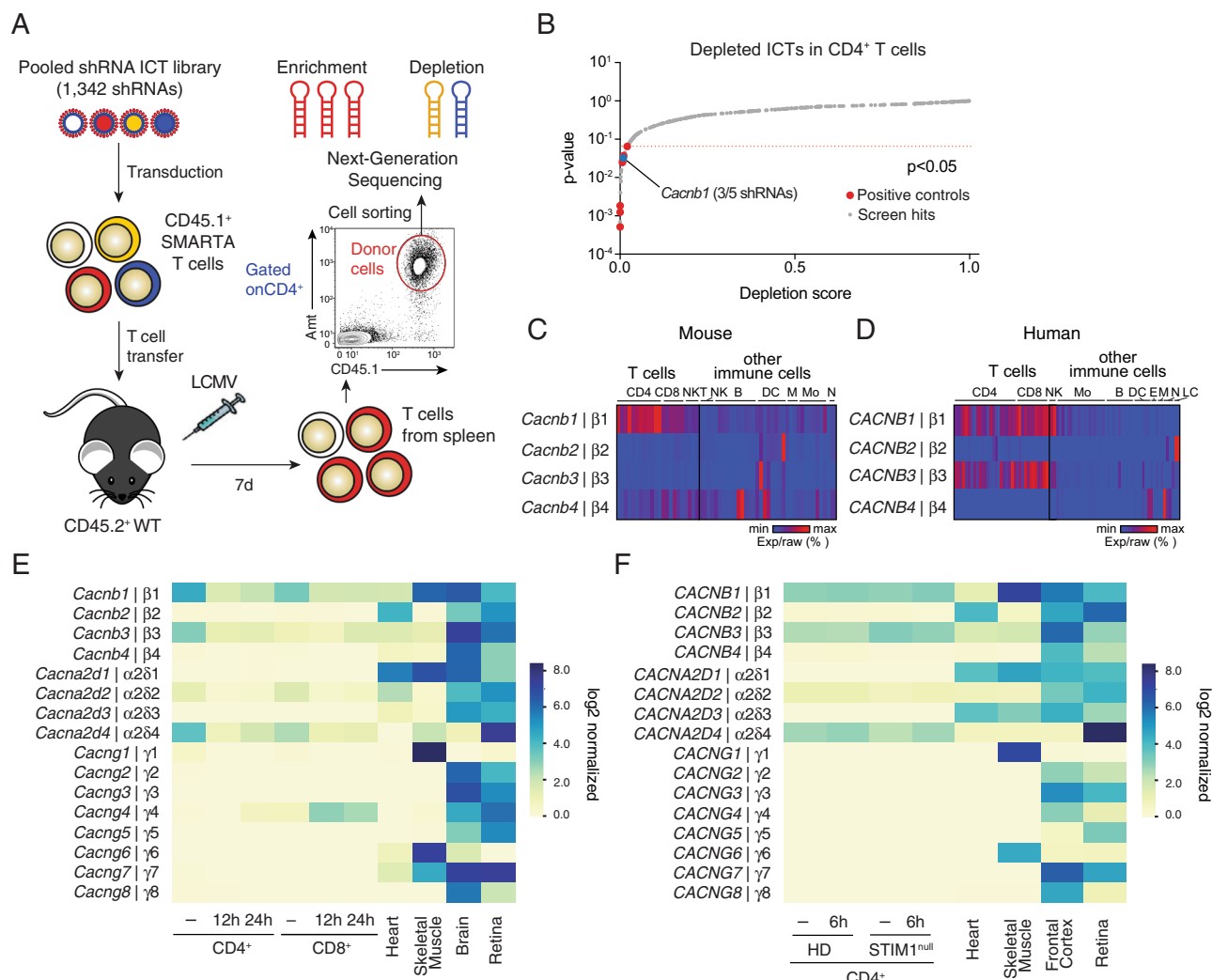

**Fig. 1 shRNA screen identifies *Cacnb1* as a regulator of antiviral T cell responses. A** In vivo ion channels and transporters (ICT) screen. CD4$^+$ T cells from SMARTA mice were transduced with a pooled shRNA library targeting 223 ICTs (1342 shRNAs including control shRNAs), enriched by cell sorting for transduced (Ametrine$^+$) cells and injected into host mice. 7 days after infection with LCMV$^{ARM}$, CD4$^+$CD45$^+$Amt$^+$ donor T cells were isolated from host spleens and analyzed by next generation sequencing (NGS) for the depletion or enrichment of shRNAs. **B** Scores of depleted shRNAs and their *p* values calculated based on the negative-binomial model using the MAGeCK software package[88]. *Cacnb1* (blue dot), other ICTs (gray) and positive controls (red) are indicated. Shown are the pooled data from three independent screens. **C, D** mRNA expression of Cavβ subunits in mouse (**C**) and human (**D**) T cells compared to other immune cells based on data from ImmGen[27] and Fantom5[28,29] databases. Each column represents a different type of immune cell. Heatmaps represent % raw min-max expression for each gene. NK natural killer, NKT natural killer T, Mo monocyte, B B cell, DC dendritic cell, E eosinophil, M macrophage, N neutrophil, LC Langerhans cell. **E, F** Absolute mRNA expression of auxiliary β, α2δ and γ subunits of VGCCs in mouse (**E**) and human (**F**) T cells and reference tissues. Mouse CD4$^+$ and CD8$^+$ T cells were left unstimulated (−) or stimulated for 12 or 24 h with anti-CD3 + CD28 antibodies. Human CD4$^+$ T cells were left unstimulated (−) or stimulated for 6 h with anti-CD3 + CD28 antibodies. mRNA expression was analyzed by RNA sequencing. HD represents the averaged data from three individual healthy donors (HD) and a patient with a STIM1 null mutation (STIM1$^{null}$). RNA-Seq for mouse T cells and for human and mouse heart, skeletal muscle, brain, frontal cortex, and retina were extracted from GEO datasets (Supplementary Table 2).

expression levels in immune cells using the Immunological Genome Project (ImmGen)[27] and Fantom5[28,29] databases, respectively (Supplementary Fig. 1A). We identified 154 ICTs that are expressed at least twofold above the population average in both mouse and human CD4$^+$ T cells (Supplementary Fig. 1B, C). Similar analyses were conducted in 11 other immune cell populations, resulting in a total of 223 ICTs with >2-fold above average expression across all cell types (Supplementary Fig. 1B). We used this information to generate a customized, pooled shRNA library targeting 223 mouse ICT genes. To delete ICTs, CD4$^+$ CD45.1$^+$ T cells were isolated from SMARTA mice that express a transgenic TCR specific for the LCMV GP$_{61-80}$

epitope[30], and transduced with the shRNA library. shRNA-transduced (Ametrine$^+$) T cells were sorted and injected into CD45.2$^+$ congenic WT host mice, which were next infected with the Armstrong strain of LCMV (LCMV$^{ARM}$, Fig. 1A). LCMV$^{ARM}$ causes an acute viral infection and a well-characterized CD4$^+$ and CD8$^+$ T cell response[31]. 7 days later, donor T cells were isolated from the spleens of host mice, enriched for CD4$^+$ CD45.1$^+$ Amt$^+$ T cells by cell sorting and analyzed by next generation sequencing (NGS) for the enrichment or depletion of ICTs targeted by specific shRNAs (Fig. 1A). Among the positive controls whose suppression resulted in the significant depletion (*P* < 0.05, Log2-FC < 0.5) of T cells 7 days after LCMV infection were genes that are critical for T

cell signaling, survival and function (*CD3e*, *Cd4*, *Lck*, *Rpa3*, *Zap70*, *Bcl2l1*). By contrast, suppression of *Prdm1*, which encodes BLIMP1 and inhibits the differentiation of follicular T helper (T_FH) cells after viral infection, resulted in significant enrichment ($P < 0.05$, Log2-FC > 2) of T cells (Fig. 1B, and Supplementary Fig. 1D).

Among the ICTs whose shRNA-mediated knockdown significantly depleted T cells in vivo was *Cacnb1*, which encodes the auxiliary Cavβ1 subunit of VGCCs (Fig. 1B, and Supplementary Fig. 1D). Given the importance of β subunits for VGCC function in excitable cells and the reported function of β2, β3 and β4 in T cells[20,23,24], we first analyzed the expression levels of all Cavβ subunits in mouse and human T cells using the ImmGen[27] and Fantom 5[28,29] gene expression databases, respectively. Compared to other immune cells, mouse T cells express higher levels of Cavβ1 (*Cacnb1*), whereas human T cells express both β1 and β3 (encoded by *CACNB1* and *CACNB3*, Fig. 1C, D). We next analyzed absolute mRNA expression levels of β subunits in mouse and human T cells compared to reference tissues with known VGCC function using our own and published RNA-Seq data. In T cells of wildtype (WT) mice and healthy human donors (HDs), β1 and β3 were the only Cavβ subunits showing robust mRNA levels (Fig. 1E, F). Whereas β1 and β3 mRNA levels in human T cells remained high after TCR stimulation, their expression decreased in mouse CD4+ and CD8+ T cells at 12–24 h after stimulation. Analysis of other auxiliary subunits of VGCCs showed that although γ subunits are generally expressed at very low levels, $α_2δ_4$ is robustly expressed in both human and mouse T cells (Fig. 1E, F). It is noteworthy that expression levels of all four β subunits and α subunits did not markedly differ between human CD4+ and CD8+ T cells (Supplementary Fig. 1E, F). The shRNA screen and expression data suggest that Cavβ1 may have a non-redundant function in T cells during LCMV infection.

**Cavβ1 is required for T cell expansion and survival in vitro and in vivo.** Because Cavβ1 is highly and selectively expressed in mouse and human T cells compared to other β subunits, and because β2, β3 and β4 were reported to regulate T cell development and function[20,23,24], we further investigated the function of Cavβ1 in T cells. To delete *Cacnb1* expression in mouse T cells we used two approaches: (1) CRISPR/Cas9 gene editing by retrovirally transducing CD4+ T cells of *Cas9^LSL*GFP *Cd4Cre* knock-in mice with small guide (sg) RNAs targeting *Cacnb1*, and (2) shRNA-mediated knockdown by transducing mouse CD4+ T cells with individual shRNAs targeting *Cacnb1*. Both approaches achieved ~50–70% reduction of Cavβ1 mRNA and protein levels (Fig. 2A, B and Supplementary Fig. 2A, B). Deletion of Cavβ1 reduced the numbers of transduced (Amt+) CD4+ T cells after TCR stimulation in vitro relative to T cells transduced with control sgRNAs or shRNAs under co-culture conditions (Fig. 2C and Supplementary Fig. 2C). No significant defects in CD4+ T cell proliferation were detectable after *Cacnb1* deletion by either sgRNAs or shRNAs (Fig. 2D and Supplementary Fig. 2D). By contrast, we observed a significant increase in apoptosis in Cavβ1-deficient CD4+ T cells compared to control T cells (Fig. 2E and Supplementary Fig. 2E) suggesting that Cavβ1 in T cells contributes to T cell survival.

We next investigated the ability of *Cacnb1*-deficient T cells to expand after infection of mice with LCMV^ARM. CD4+ T cells from *Cas9^LSL*GFP *Cd4Cre* knock-in mice that had been crossed to SMARTA mice were transduced with sgRNAs against *Cacnb1* (Ametrine+) and mixed at a 1:1 ratio with CD4+ T cells transduced with control sgRNAs (GFP+), allowing us to investigate the effects of Cavβ1 deletion in CD4+ T cells compared to mock-transduced T cells in the same host mice. As an additional control, CD4+ T cells were transduced with control sgRNAs encoded by vectors expressing either Ametrine or GFP reporters that were also mixed at a 1:1 ratio (sgCtrl^Amt/sgCtrl^GFP). Following T cell transfer WT host mice were infected with LCMV^ARM (Fig. 2F). 7 days post-infection, we observed a significant ~2.4-fold decrease of CD4+ T cells transduced with sg*Cacnb1* (Amt+) compared to sg*Ctrl* (Amt+) when normalized to sg*Ctrl* (GFP+) transduced cells (Fig. 2F). Similar observations were made using an orthogonal approach by transducing CD4+ T cells from SMARTA mice with sh*Cacnb1* (Amt+) and sh*Ctrl* (GFP+) followed by transfer of T cells at a 1:1 ratio and infection with LCMV^ARM (Supplementary Fig. 2F). Whereas the ratio of sh*Ctrl* (Amt+) to sh*Ctrl* (GFP+) transduced CD4+ T cells remained unchanged 7 days after infection, we found a strong, ~sixfold reduction in the ratio of sh*Cacnb1* (Amt+) to shCtrl (GFP+)-transduced T cells (Supplementary Fig. 2F). Together, these data indicate that Cavβ1 is required for the clonal expansion of CD4+ T cells in vitro and in vivo after viral infection by regulating T cell survival.

**Cavβ1 expression in T cells is dispensable for TCR-mediated Ca²⁺ influx and cytokine production.** The major canonical function of Cavβ subunits is to regulate the function of VGCCs and thereby $Ca^{2+}$ signaling in excitable cells[7]. $Ca^{2+}$ signals are critical for many T cell functions including cell proliferation, survival and cytokine production[1,2]. Because previous studies had shown that Cavβ3 and β4 subunits regulate $Ca^{2+}$ influx in T cells and T cell function[23,24], we next investigated the effects of Cavβ1 deletion on $Ca^{2+}$ signaling and T cell function. Following deletion of Cavβ1 by transducing CD4+ T cells with sg*Cacnb1* or sh*Cacnb1*, T cells were stimulated by CD3 crosslinking and analyzed for cytosolic $Ca^{2+}$ concentrations. Cavβ1 deletion in T cells by either sgRNA or shRNA did not impair TCR-induced $Ca^{2+}$ influx (Fig. 3A, B and Supplementary Fig. 3A, B). Likewise, SOCE induced by 1 µM ionomycin (to deplete ER $Ca^{2+}$ stores) was not affected by suppression of *Cacnb1* expression. By contrast, deletion of *Stim1* to suppress CRAC channel activation strongly suppressed TCR-mediated $Ca^{2+}$ influx. We next analyzed if the expression of cytokines that are known to be regulated by $Ca^{2+}$ is impaired in *Cacnb1*-deficient T cells. Deletion of Cavβ1 in CD4+ T cells by transduction with sgRNAs and shRNAs had no effect on the production of IL-2, TNF and IFN-γ in response to PMA/ionomycin stimulation compared to T cells transduced with control sgRNAs and shRNAs (Fig. 3C, D and Supplementary Fig. 3C, D). By contrast, deletion of STIM1-strongly suppressed the production of all three cytokines. Collectively, these data demonstrate that Cavβ1 is dispensable for TCR-induced $Ca^{2+}$ signaling and cytokine production in T cells.

**Depolarization does not evoke Ca²⁺ influx or Ca²⁺ currents in T cells.** Whether VGCCs are functional as $Ca^{2+}$ channels in T cells and regulate $Ca^{2+}$ signaling has remained controversial. The normal $Ca^{2+}$ signals in Cavβ1-deficient T cells despite altered T cell function prompted us to investigate whether VGCC function is detectable in human and mouse T cells. To this end, we measured $Ca^{2+}$ signals in T cells following exposure to high extracellular concentrations of K+ ($[K^+]_o$) to depolarize the membrane potential ($V_m$) as was first demonstrated in lymphocytes by Deutsch et al.[32] and shown to activate VGCCs in excitable cells[33]. 60 mM and 150 mM $[K^+]_o$ are predicted (using the Goldman–Hodgkin–Katz equation) to depolarize the $V_m$ of T cells from ~−55 mV to −24 mV and 0 mV, respectively. Exposure to 60 or 150 mM $[K^+]_o$ did not induce an increase in intracellular $[Ca^{2+}]$ in mouse (Fig. 4A) or human CD4+ T cells (Fig. 4B). By contrast, depletion of ER $Ca^{2+}$ stores with ionomycin induced robust SOCE in mouse and human T cells at

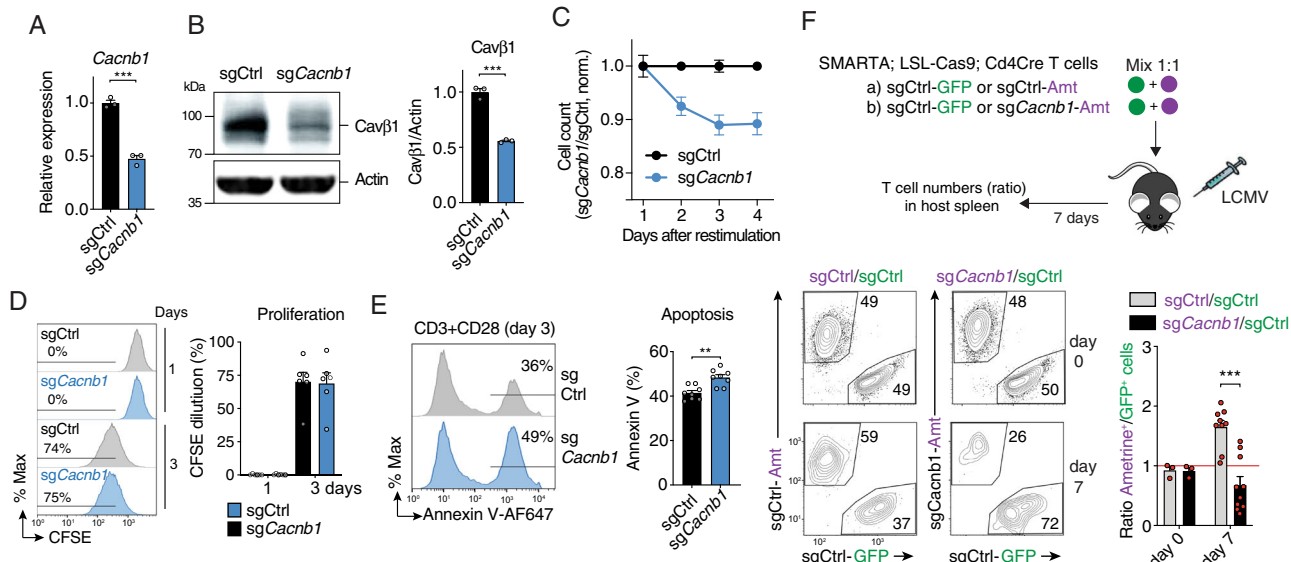

**Fig. 2 Deletion of Cavβ1 impairs viability of CD4+ T cells and their expansion after viral infection. A** mRNA expression of *Cacnb1* in CD4+ T cells of LSL-Cas9; Cd4Cre mice transduced with control sgRNA (sgCtrl) and sgRNA targeting *Cacnb1*. mRNA levels were measured in transduced (Ametrine+) T cells by qPCR at day 3 post-transduction. *Rlp32* was used as housekeeping control. sg*Cacnb1* samples were normalized to sgCtrl. **B** Representative Western blot (left) and quantification (right) of Cavβ1 protein in CD4+ T cells transduced with sgCtrl or sg*Cacnb1*. After 4–5 days, Cavβ1 was detected using a monoclonal antibody recognizing aa 19-34 in the N-terminus of Cavβ1. Data in (**A**, **B**) are the mean ± SEM of $n = 3$ mice from independent experiments. **C**–**E** CD4+ T cells from LSL-Cas9; Cd4Cre mice were transduced with sgCtrl or sg*Cacnb1* and at day 4 restimulated with anti-CD3 + CD28. **C** Cell counts shown as the ratio of sg*Cacnb1* / sgCtrl transduced T cells normalized to non-transduced T cells. **D** Representative flow cytometry plots (left) and quantification (right) of CFSE dilution at 1 and 3 days after re-stimulation. **E** Representative flow cytometry plots (left) and quantification (right) of apoptosis measured by annexin V staining at 3 days after re-stimulation. Data in (**C**–**E**) are the mean ± SEM of $n = 6$ mice (in **C**, **D**) and $n = 8$ mice (in **E**). **F** Adoptive transfer of CD4+ T cells from SMARTA LSL-Cas9; Cd4Cre mice that has been transduced with sgCtrl or sg*Cacnb1* followed by LCMV[ARM] infection. Transduced donor T cells were mixed at 1:1 ratio before injection. At day 7 post-infection, the ratios of sg*Cacnb1*/sgCtrl T cells (and sgCtrl/sgCtrl) were analyzed. Representative flow cytometry plots (bottom left) and quantification (bottom right) of T cell ratios. Data are the mean ± SEM from $n = 3$ independent experiments pooled from $n = 3$ donor SMARTA; LSL-Cas9; Cd4Cre mice and $n = 10$ WT host mice per group. Statistical analysis was conducted by two-tailed, unpaired Student's *t* test. **p < 0.01, ***p < 0.001.

physiological extracellular $[K^+]_o$ (4.5 mM), which was suppressed by depolarization of $V_m$ in 60 mM $[K^+]_o$ (Fig. 4B). This reduction is expected because depolarization of $V_m$ collapses the electrical gradient required for $Ca^{2+}$ influx through store-operated CRAC channels. To demonstrate that depolarization of $V_m$ by application of high extracellular $K^+$ is able to activate VGCCs, we transfected HEK293 cells, which are not excitable, with the α1 subunit of Cav1.2 together with β, γ and α2δ subunits and subjected these cells to the same depolarization protocol. Addition of 150 mM extracellular $K^+$ evoked $Ca^{2+}$ influx in Cav1.2 transfected cells, but not in untransfected cells (Supplementary Fig. 4A). As expected, $Ca^{2+}$ influx in Cav1.2 expressing cells could be blocked by the $Ca^{2+}$ channel blocker nimodipine (8 μM) (Supplementary Fig. 4B). Moreover, we tested if depolarization of $V_m$ by high extracellular $K^+$ induces voltage-dependent $Ca^{2+}$ influx in PC12 cells which endogenously express VGCCs[34]. Exposure of PC12 cells to 150 mM $[K^+]_o$ induced a robust, transient increase in intracellular $Ca^{2+}$ levels (Supplementary Fig. 4C).

To more precisely and dynamically control the membrane potential in T cells and to measure voltage-dependent $Ca^{2+}$ influx and $Ca^{2+}$ currents in T cells, we investigated VGCC channel function by patch-clamp electrophysiology. Human T cells from healthy donors (HD) were loaded with the $Ca^{2+}$ sensitive dye Indo-1 to measure $[Ca^{2+}]_i$ concentrations and patch-clamped to record VGCC currents. The perforated-patch configuration was chosen to minimize run-down of VGCC currents that commonly occurs during whole-cell recordings. Simultaneous measurements of $[Ca^{2+}]_i$ and VGCC currents provided two independent ways to

measure depolarization-evoked $Ca^{2+}$ signals. In separate experiments, we measured VGCC currents in the presence of 110 mM $Ba^{2+}$ as the charge carrier, which conducts through the channel about twofold better than $Ca^{2+}$ and confers the added advantage that $Ba^{2+}$ currents, unlike $Ca^{2+}$ currents, do not inactivate, thereby optimizing detection of even small VGCC currents in T cells. To activate VGCCs, human T cells of HD were depolarized stepwise from −60 to +60 mV from a holding potential of −80 mV. No inward currents were detectable with this protocol, either in isotonic $Ba^{2+}$ (Fig. 4C) or in 20 mM $Ca^{2+}$ (Fig. 4D). Furthermore, steps to various voltages between −60 and +60 mV failed to evoke increases in $[Ca^{2+}]_i$ in human T cells (Fig. 4D). Although this result strongly suggests that human T cells do not express functional VGCCs, this conclusion may not be valid if T cells only expressed a few functional VGCCs per cell and the resulting small current amplitudes and $Ca^{2+}$ signals were hard to detect. To circumvent this potential limitation, we repeatedly activated VGCCs by delivering depolarizing stimuli every second with the goal of eliciting a more pronounced rise in $[Ca^{2+}]_i$. We applied 40–50 depolarizing steps (each lasting 200 ms) to +10 mV from the holding potential every second. This protocol also failed to elicit a rise in $[Ca^{2+}]_i$ in human T cells (Fig. 4E). To exclude the possibility that the sensitivity of our recording system is too low to record small $Ca^{2+}$ currents, we measured $Ca^{2+}$ influx resulting from the activation of CRAC channels on the same experimental set-up. After treatment with thapsigargin to deplete $Ca^{2+}$ stores and activate CRAC channels, human T cells were held at a positive membrane potential of +60 mV and repeatedly stimulated with voltage steps to

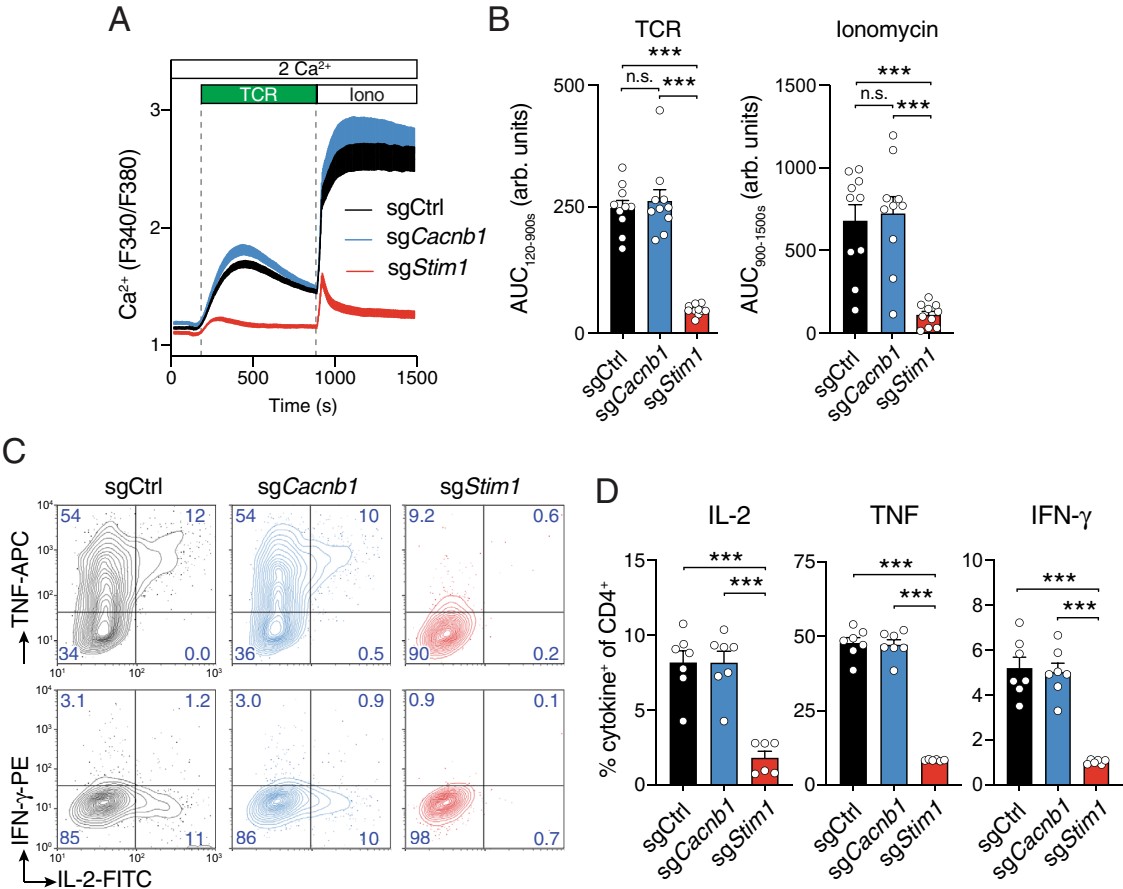

**Fig. 3 Cavβ1 is not required for Ca²⁺ influx and cytokine production by T cells.** CD4⁺ T cells of LSL-Cas9; Cd4Cre mice were transduced with sgCtrl, sgCacnb1 or sgStim1. **A, B** After 3 days, Amt⁺ T cells were enriched by cell sorting, recovered for one day in medium containing IL-2 and IL-7 and analyzed. Cytosolic Ca²⁺ levels were measured following stimulation of T cells by anti-CD3 (TCR) cross-linking and ionomycin (Iono) in Ringer's solution containing 2 mM Ca²⁺. Averaged Ca²⁺ traces (**A**) and quantification (**B**) of the area under the curve (AUC) in the time periods indicated by the dotted lines. **C, D** Cytokine production by CD4⁺ T cells was measured at day 4 after transduction and restimulation for 6 h with PMA and ionomycin. Representative contour plots (**C**) and quantification (**D**) of IL-2⁺, TNF⁺ and IFN-γ⁺ CD4⁺ T cells. Data in (**A, B, D**) are the mean ± SEM of $n = 7$ independent experiments performed in duplicates. Statistical analysis by two-tailed, unpaired Student's $t$ test. ***$p < 0.001$.

−100 mV delivered every 1 s. Under these conditions we observed a robust rise in $[Ca^{2+}]_i$ that peaked at ~500 nM (Fig. 4F, H). This rise in $[Ca^{2+}]_i$ was paralleled by Ca²⁺ currents with an inwardly rectifying current-voltage relationship typical of CRAC channels (Fig. 4G, H). In these latter experiments, a 6 nM rise in $[Ca^{2+}]_i$ immediately following readdition of extracellular Ca²⁺ could be detected with the 170 fC of Ca²⁺ influx that flowed through CRAC channels (assessed by integrating the Ca²⁺ current charge over the 200 ms duration of the step-ramp pulse) (Fig. 4G), demonstrating that the inability to observe increases in $[Ca^{2+}]_i$ after depolarization of T cells was not due to a low sensitivity of our recording system. To further exclude the possibility that recording conditions are not sensitive enough to detect VGCC currents, we analyzed voltage-gated Ca²⁺ currents in HEK293 cells transfected with Cav1.2 and PC12 cells endogenously expressing VGCCs using the whole-cell patch clamp configuration. Stepwise depolarization of HEK293 cells transfected with Cav1.2 channels from −80 mV to +80 mV evoked robust voltage-gated Ca²⁺ currents that could be blocked completely with 8 μM nimodipine (Supplementary Fig. 5A). Similarly, applying depolarizing steps in PC12 cells from −80 mV to +60 mV induced obvious voltage-gated Ca²⁺ currents in these neuroendocrine cells (Supplementary Fig. 5B). Collectively, these experiments demonstrate that even under sensitive recording conditions neither voltage-activated Ca²⁺ influx nor Ca²⁺

currents consistent with the presence of functional VGCCs can be detected in human T cells.

**TCR stimulation fails to evoke depolarization-induced Ca²⁺ influx and VGCC currents in mouse and human T cells.** Although membrane depolarization is sufficient to activate VGCCs and Ca²⁺ influx in excitable cells, we hypothesized that in non-excitable T cells an additional stimulus may be required to activate VGCCs. To test this hypothesis, we measured Ca²⁺ influx in mouse CD4⁺ T cells that were stimulated by CD3-crosslinking prior to depolarization with high $[K^+]_o$. TCR stimulation in the presence of physiological $[K^+]_o$ resulted in Ca²⁺ influx, which could be further amplified by inducing SOCE with ionomycin (Fig. 5A). TCR stimulation followed by depolarization with 60 or 150 mM $[K^+]_o$ did not increase Ca²⁺ influx; instead 150 mM $[K^+]_o$ significantly reduced $[Ca^{2+}]_i$ (Fig. 5A, B). Likewise, SOCE induced by ionomycin stimulation was decreased by 150 mM $[K^+]_o$. Similar results were observed in human T cells that were stimulated with anti-CD3 antibody (OKT3) after membrane depolarization with either 60 mM or 150 mM $[K^+]_o$ (Supplementary Fig. 6A, B). The fact that membrane depolarization decreases, rather than increases, Ca²⁺ signals in T cells is consistent with the requirement for a negative $V_m$ to provide the electrical gradient for Ca²⁺ influx through CRAC and other Ca²⁺ channels.

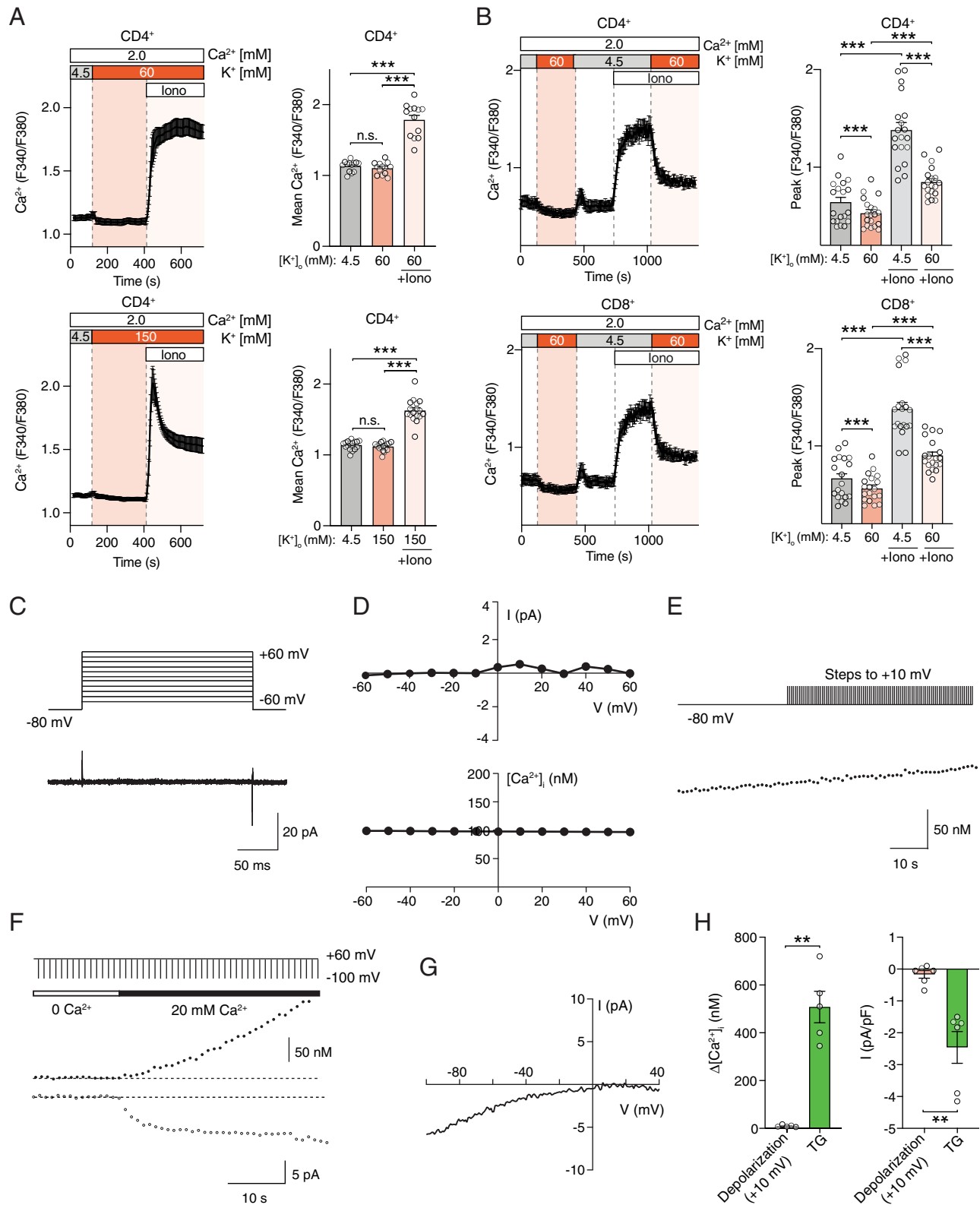

To directly evaluate whether functional VGCC channels are present in mouse T cells, we carried out measurements of whole-cell $Ca^{2+}$ currents using patch-clamp electrophysiology in response to depolarizing voltage steps. $CD4^{+}$ T cells isolated from the spleen of C57BL/6 mice were kept overnight in culture medium supplemented with IL-7 or stimulated for 2 days with anti-CD3/CD28 antibodies before analysis of channel function.

To activate VGCCs, T cells were depolarized stepwise from −80 to +60 mV from a holding potential of −70 mV. These recordings failed to induce any inward currents in naïve unstimulated or stimulated mouse T cells (Fig. 5C, D). As a control for our recording conditions, we measured $K^{+}$ currents resulting from the voltage dependent activation of Kv1.3, which is well known to regulate the membrane potential of T cells and T

**Fig. 4 Depolarization of T cells fails to evoke Ca²⁺ influx and Ca²⁺ currents.** **A** Cytosolic $Ca^{2+}$ levels in mouse $CD4^+$ T cells. T cells were stimulated with anti-CD3 + CD28, cultured for 3 days and exposed to 60 mM (top) and 150 mM (bottom) KCl followed by stimulation with ionomycin (Iono). **B** Cytosolic $Ca^{2+}$ levels in human T cells from a healthy donor (HD) cultured for 10 days in vitro, exposed to 60 mM KCl and stimulated with ionomycin. Averaged $Ca^{2+}$ traces (left) and quantification (right) of the mean F340/F380 ratio during the time periods indicated by dotted lines. Data shown are the mean ± SEM of $n = 3$–4 (in **A**) and $n = 7$ (in **B**) independent experiments conducted in duplicates. **C–H** No voltage-gated $Ca^{2+}$ currents and signals in human T cells. **C** Membrane currents in HD T cells were recorded in 110 mM $Ba^{2+}$ in response to voltage steps from −60 to +60 mV for 200 ms from a holding potential of −80 mV. Current traces were leak-subtracted using the P/8 method with steps from −100 mV. **D** Current-voltage (I–V) plots (top) and $[Ca^{2+}]_i$ concentrations (bottom) measured using Indo-1 in the same cell stimulated in the presence of 20 mM extracellular $Ca^{2+}$ using the voltage protocol shown in (**C**). **E** $[Ca^{2+}]_i$ was measured in Indo-1 loaded HD T cells held at −80 mV for 20–30 s to establish the baseline $[Ca^{2+}]_i$ followed by application of 40-50 depolarizing steps to +10 mV every 1 s. **F, G** Simultaneous measurements of $[Ca^{2+}]_i$ and $I_{CRAC}$. **F** T cells pretreated with TG were exposed to a step-ramp voltage protocol comprising a −100 mV step (50 ms) followed by a ramp from −100 to +100 mV (50 ms) every 1 s. The holding potential was +60 mV to prevent $Ca^{2+}$ influx during the interpulse interval. **G** Representative I–V plot typical of $I_{CRAC}$ recorded during the −100 mV pulse from the experiments shown in (**F**). **H** $[Ca^{2+}]_i$ rises (left) and current densities (right) in response to either depolarizing steps (+10 mV) or TG treatment. For details see Methods. Data in (**C–H**) represent the mean ± SEM from $n = 5$–6 cells per condition. Statistical analysis by two-tailed, unpaired Student's $t$ test. **$p < 0.01$, ***$p < 0.001$.

cell function[35]. Stepwise depolarization of unstimulated and stimulated $CD4^+$ T cells from −100 to +100 mV evoked robust, easily detectable voltage-activated outwardly rectifying $K^+$ currents whose voltage dependence and kinetics were similar to well-described Kv1.3 channels (Fig. 5E, F)[36].

We next attempted to detect VGCC currents in human T cells after acute TCR stimulation with OKT3 (Fig. 5G, H). Examination of voltage-gated currents in T cells from HDs is complicated by the fact that TCR stimulation triggers the activation of CRAC currents and a rise in $[Ca^{2+}]_i$. The presence of CRAC currents precluded us from employing leak subtraction using the *P/N* method[37] and VGCC currents were therefore measured without leak subtraction. Instead, we tested whether currents observed during T cell depolarization were affected by nimodipine, a potent blocker of L-type VGCCs. Stepwise depolarization of OKT3-treated human T cells failed to evoke inward $Ba^{2+}$ currents resembling VGCC currents. Furthermore, application of 8 μM nimodipine failed to affect the observed membrane currents (Fig. 5G, H), suggesting that dihydropyridine (DHP)-sensitive currents are not present in stimulated human T cells. Together, our data indicate that VGCCs do not contribute to $Ca^{2+}$ signals in T cells in response to TCR stimulation.

In neurons, the function of L-type VGCCs is modulated by the phosphorylation of $\alpha_1$ subunits mediated by protein kinase A (PKA)[38]. Activation of PKA by cAMP downstream of β-adrenergic receptor stimulation results in increased L-type $Ca^{2+}$ currents[3]. Protein kinase C (PKC) also regulates L-type $Ca^{2+}$ channel function and has both stimulatory and inhibitory effects on $Ca^{2+}$ currents[39]. PKA and PKC also mediate T cell signaling and control T cell function[40,41]. To test the hypothesis that PKA or PKC may be required for VGCC function in T cells, we preincubated human T cells with the PKC agonist phorbol 12-myristate 13-acetate (PMA) or the PKA agonist 8-Bromoadenosine 3′,5′-cyclic monophosphate (8-Br-cAMP) for 10 min before depolarizing cells with 60 mM $[K^+]_o$. Neither treatment with PMA nor 8-Br-cAMP induced $Ca^{2+}$ influx upon depolarization (Supplementary Fig. 7A–D). Collectively, these data demonstrate that neither voltage-gated $Ca^{2+}$ influx nor $Ca^{2+}$ currents are detectable in human or mouse T cells, regardless if T cells were left unstimulated or restimulated by TCR activation or direct activation of PKC or PKA signaling.

**Lack of ORAI1, STIM1 and CRAC channel function does not induce compensatory VGCC currents and Ca²⁺ influx in T cells.** The contribution of VGCCs to TCR-induced $Ca^{2+}$ influx may be difficult to assess in WT T cells because TCR stimulation results in the activation of CRAC channels and SOCE, which could mask smaller $Ca^{2+}$ signals emanating from VGCCs. To circumvent

this problem, we investigated whether VGCC currents are unmasked in the absence of functional CRAC channels. To this end, we used T cells from a patient with a loss-of-function (LOF) mutation in *ORAI1* (p.R91W), which abolishes CRAC channel function and SOCE[42]. We activated VGCCs in ORAI1-deficient T cells by depolarizing the membrane stepwise from −60 to +60 mV from a holding potential of −80 mV using the perforated patch configuration and simultaneously measured $[Ca^{2+}]_i$ with Indo-1 (Fig. 6A). No inward $Ca^{2+}$ currents or depolarization-evoked $Ca^{2+}$ signals could be detected with this protocol using either external solutions containing 20 mM $Ca^{2+}$ (Fig. 6B). To again exclude the possibility that activation of VGCCs in T cells requires TCR signaling in addition to depolarization, ORAI1-deficient T cells were first stimulated with OKT3 followed by stepwise depolarization. TCR cross-linking in the absence of CRAC channels failed to unmask depolarization-evoked membrane currents and $Ca^{2+}$ signals (Fig. 6C). We reasoned again that we might not be able to detect small VGCC currents if T cells expressed only a few functional VGCCs per cell. We therefore measured $[Ca^{2+}]_i$ in Indo-1-loaded T cells that were repeatedly depolarized every 1 s to produce a buildup of cytosolic $Ca^{2+}$. Neither ORAI1-deficient T cells that were left untreated nor cells stimulated with OKT3 showed increases in $[Ca^{2+}]_i$ upon repeated depolarization (Fig. 6D, E). By contrast, stimulation of ORAI1-deficient T cells with a high dose of ionomycin to bypass $Ca^{2+}$ influx through CRAC channel resulted in the robust elevation of $[Ca^{2+}]_i$ (Fig. 6F). Collectively, these data demonstrate that the absence of functional CRAC channels does not result in compensatory activity of VGCCs.

Previous reports have shown that activation of STIM1, which is essential for the activation of CRAC channels by binding to ORAI[43], inhibits $Ca^{2+}$ influx through L-type Cav1.2 channels in response to depolarization[44,45]. These studies suggested that STIM1 reciprocally activates CRAC channels while suppressing Cav1.2. To test the hypothesis that STIM1 suppresses VGCC function in T cells, we first analyzed $Ca^{2+}$ influx in T cells from a patient with a null mutation in *STIM1* (c.497 + 776 A > G) that abolishes STIM1 protein expression. Application of 60 mM $[K^+]_o$ to depolarize $V_m$ cells did not evoke a rise in $[Ca^{2+}]_i$ in T cells from either a HD or the STIM1-deficient patient (Fig. 6G). Whereas ionomycin induced SOCE in HD T cells, no increase in $[Ca^{2+}]_i$ was observed in the absence of STIM1 (Fig. 6G). We next analyzed $Ca^{2+}$ signals in T cells from WT and *Stim1^{fl/fl}Cd4Cre* mice with conditional deletion of STIM1 in T cells. Depolarization of *Stim1*-deficient mouse T cells by application of 60 mM or 150 mM $[K^+]_o$ failed to induce a rise in $[Ca^{2+}]_i$ (Fig. 6H, I). In WT T cells, depolarization with high $[K^+]_o$ suppressed SOCE induced by ionomycin stimulation (Fig. 6H, I). To exclude the possibility that VGCC function in

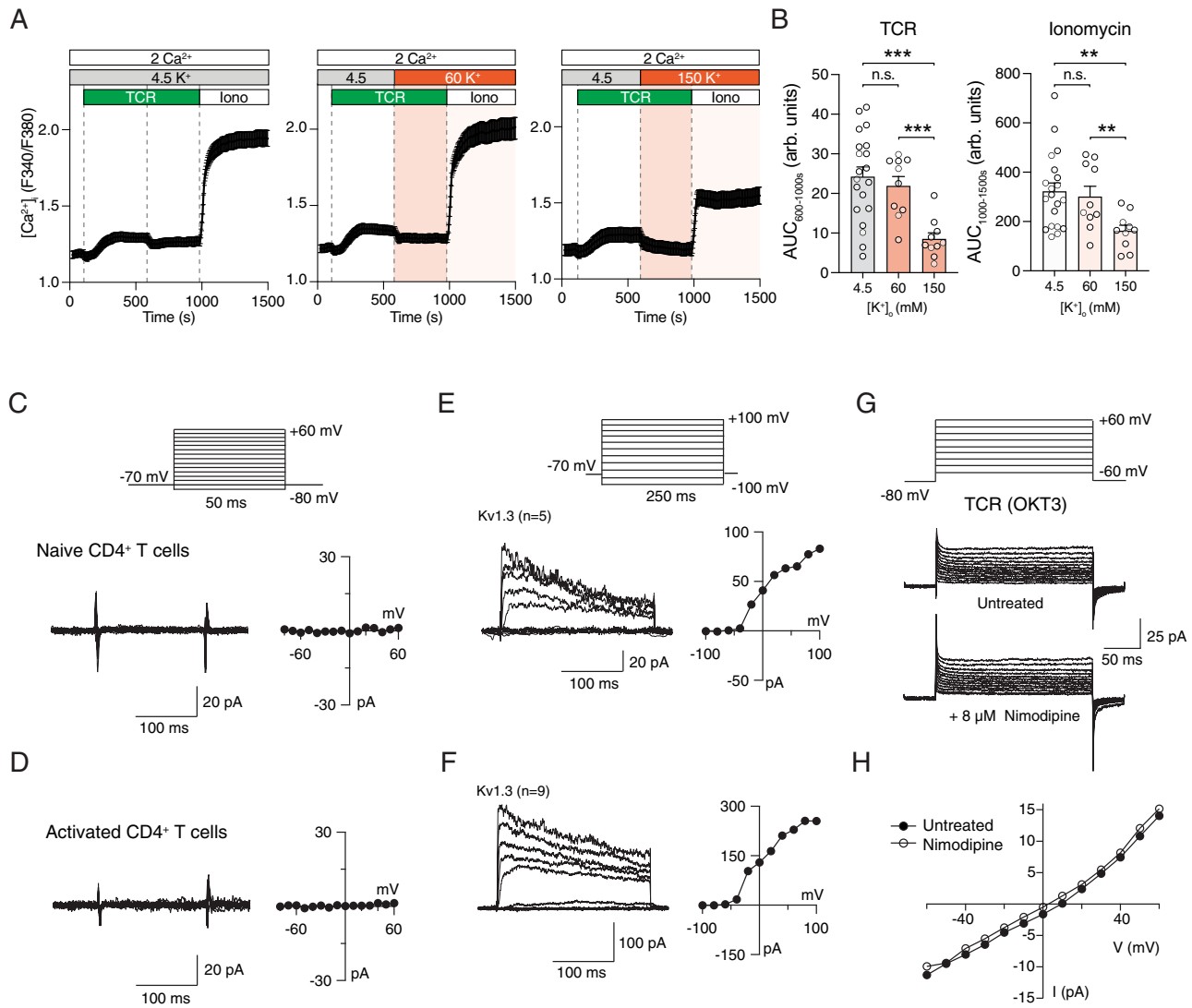

**Fig. 5 Lack of voltage-dependent Ca$^{2+}$ influx and Ca$^{2+}$ current in T cells after simultaneous TCR stimulation. A, B** Cytosolic Ca$^{2+}$ levels in CD4$^+$ T cells isolated from WT C57BL/6 mice. T cells were stimulated by anti-CD3 (TCR) crosslinking followed at 600 sec by exposure to Ringer's solution containing either 4.5 mM (left), 60 mM (middle) or 150 mM (right) KCl. Stimulation with ionomycin (Iono) at 1000 s was used as positive control. **A** Averaged Ca$^{2+}$ traces and (**B**) quantification of the area under the curve (AUC) during the indicated recording periods. Data represent the mean ± SEM of $n = 5$ experiments conducted in duplicates. **C, D** No voltage-activated Ca$^{2+}$ currents are detectable in naïve (**C**) and expanded (**D**) mouse T cells. The membrane voltage was stepped from −80 to +60 mV in increments of 10 mV for 50 ms from a holding potential of −70 mV. Currents were leak-subtracted by collecting traces for the same voltage steps in the presence of 100 µM LaCl$_3$. The current-voltage plot is shown on the right in each case. **E, F** Voltage-activated K$^+$ currents (Kv1.3) in naïve (**E**) and activated (**F**) mouse T cells. Kv currents were elicited by depolarizing steps from −100 and +100 mV in increments of 20 mV from a holding potential of −70 mV. The right plot shows the I–V relationship of the recorded Kv currents. Currents were leak-subtracted using the P/8 method. **G, H** Depolarization fails to induce voltage-gated Ca$^{2+}$ current in activated T cells. **G** Human T cells from a healthy donor (HD) were stimulated with anti-CD3 antibodies (OKT3) and membrane currents were recorded in extracellular Ringer's solution containing 110 mM Ba$^{2+}$. Displayed are the raw current traces (without leak subtraction) in response to depolarizing voltage stimuli from −60 to +60 mV in steps of 10 mV for 200 ms from a holding potential of −80 mV. To identify VGCC currents, OKT3-stimulated HD T cells were left untreated or treated with 8 µM nimodipine. **H** I–V plot of cells shown in (**G**) in the absence (black circles) or presence (open circles) of nimodipine. Data in (**C–H**) are representative of the following number of cells analyzed: $n = 16$ (**C**), $n = 14$ (**D**), $n = 5$ (**E**), $n = 9$ (**F**), $n = 3$ (**G, H**). Statistical analysis in (**B**) by two-tailed, unpaired Student's $t$ test. **$p < 0.01$, ***$p < 0.001$.

T cells in the absence of STIM1 requires both TCR stimulation and depolarization, we stimulated T cells from a HD and the STIM1-deficient patient with OKT3 after depolarization with 60 mM or 150 mM [K$^+$]$_o$. Depolarization suppressed the TCR induced increase in [Ca$^{2+}$]$_i$ observed in HD T cells, and failed to evoke a rise in [Ca$^{2+}$]$_i$ in STIM1-deficient T cells (Supplementary Fig. 8A, B). Similar observations were made in mouse T cells from WT and *Stim1$^{fl/fl}$ Cd4Cre* mice. TCR crosslinking followed by depolarization with 60 mM or 150 mM [K$^+$]$_o$ did

not evoke an increase in [Ca$^{2+}$]$_i$ in STIM1-deficient T cells (Supplementary Fig. 8C, D). In WT T cells, depolarization suppressed Ca$^{2+}$ influx following TCR and ionomycin stimulation as expected. Collectively, these data demonstrate that STIM1 deletion in either human or mouse T cells fails to induce voltage-gated Ca$^{2+}$ channel activity.

**Several α1 pore subunits of VGCCs are expressed in T cells but lack the N terminus.** Several studies have reported mRNA and/or

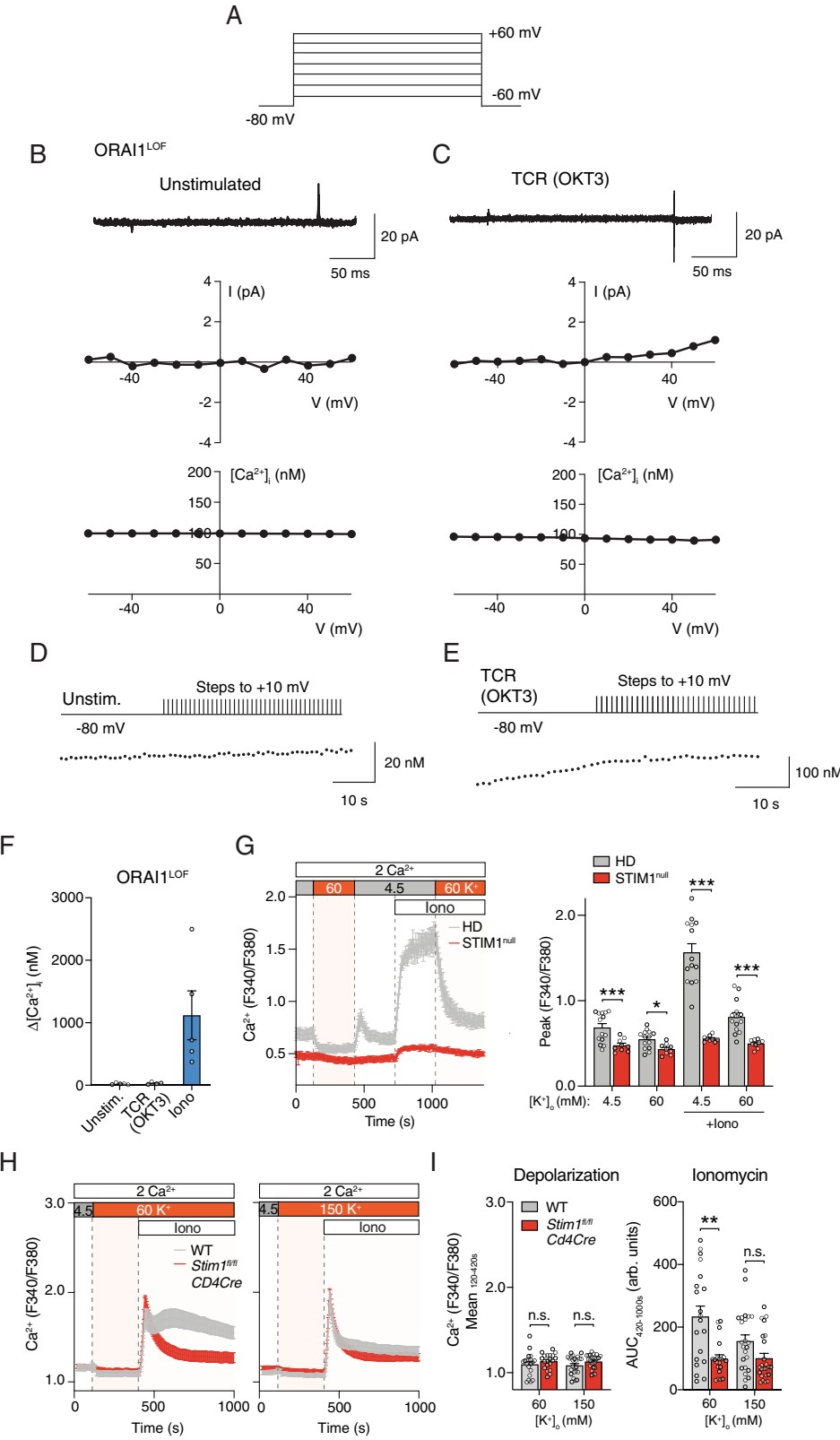

protein expression in T cells of L- and T-type VGCCs including Cav1.2, Cav1.3, Cav1.4 and Cav3.1[17,19,46]. Because we were unable to detect evidence for the presence of functional VGCCs in T cells, this raises the question if the α1 pore subunits of VGCCs are expressed in human and mouse T cells. To address this question, we first investigated the protein levels of the L-type Ca²⁺ channels Cav1.2, Cav1.3 and Cav1.4, which had previously been reported in

T cells[15,17]. For Cav1.2, no protein band was detectable in lysates of human T cells and only a very weak band in mouse T cells (Supplementary Fig. 9A) suggesting nearly complete or complete absence of Cav1.2 in T cells. The specificity of the anti-Cav1.2 antibody used was confirmed using HEK293 cells transfected with either Cav1.2 or Cav1.3. Some putative Cav1.3 protein expression was observed in mouse, but not human, T cells although the identity of the observed

**Fig. 6 Lack of ORAI1 or STIM1 does not induce voltage-gated Ca$^{2+}$ current or Ca$^{2+}$ influx in T cells. A–C** Perforated patch recordings of human T cells from a patient with a loss-of-function (LOF) mutation (p.R91W) in ORAI1 (ORAI1$^{LOF}$). T cells were left untreated (**B**) or stimulated with OKT3 (**C**) for 5–25 min prior to measurements. To record Ca$^{2+}$ current and cytosolic Ca$^{2+}$ levels, T cells were stepped from −60 to +60 mV for 200 ms from a holding potential of −80 mV. Displayed are membrane currents (top), I–V plots (middle) and [Ca$^{2+}$]$_i$ traces (bottom) measured simultaneously in the same cells in 20 mM Ca$^{2+}$ solution. Currents were leak-subtracted using the *P/8* method. Data shown in (**B**, **C**) are representative of $n = 7$ and $n = 3$ cells, respectively. **D, E** T cells of the ORAI1$^{LOF}$ patient were loaded with Indo-1 and either left unstimulated (**D**) or stimulated with OKT3 (**E**). T cells were stepped to +10 mV for 200 ms every second from a holding potential of −80 mV. Ca$^{2+}$ traces in (**D**, **E**) are representative of $n = 5$ and $n = 4$ cells, respectively. **F** Quantification of [Ca$^{2+}$]$_i$ at +10 mV in unstimulated and OKT3-stimulated T cells and after treatment with 5 µM ionomycin (Iono). $\Delta$[Ca$^{2+}$]$_i$ was measured as the difference between the [Ca$^{2+}$]$_i$ prior to and at the end of 30 depolarization pulses. Data shown are the mean ± SEM from $n = 4$–5 cells. **G** Cytosolic Ca$^{2+}$ levels in human T cells from a patient with a STIM1 c.497 + 776 A > G null mutation (STIM1$^{null}$). T cells were cultured for 10 days in vitro, loaded with Fura-2 and depolarized with Ringer's solution containing 60 mM KCl. Averaged Ca$^{2+}$ traces (left) and quantification (right) of the peak F340/F380 ratios during the time periods indicated by dotted lines. Data are the mean ± SEM from $n = 5$ independent experiments. (Note that Ca$^{2+}$ traces of the HD T cells are the same as those shown in Fig. 4B; HD T cells were analyzed together with STIM1$^{null}$ T cells and are shown for comparison). **H, I** Cytosolic Ca$^{2+}$ levels in CD4$^+$ T cells from wildtype (WT) and *Stim1$^{fl/fl}$Cd4Cre* mice. T cells were activated for 3 days with anti-CD3/CD28 and then depolarized with Ringer's solution containing 60 mM or 150 mM KCl. **H** Averaged Ca$^{2+}$ traces and (**I**) quantification of the mean (left) and AUC (right) of F340/F380 ratios during the indicated time periods. Data represent the mean ± SEM from $n = 5$ independent experiments. Statistical analysis by two-tailed Mann–Whitney U test. *$p < 0.05$, **$p < 0.01$, ***$p < 0.001$.

bands at the expected 230 kDa size is not certain in part because they migrated slightly faster than the reference band in HEK293 cells overexpressing Cav1.3 and because we also detected an (albeit weaker) band in HEK293 cells transfected with Cav1.2 (Supplementary Fig. 9A). Cav1.4 protein was readily detectable in mouse retina, but was absent in thymocytes (Supplementary Fig. 9B). The specificity of the anti-Cav1.4 antibody was confirmed using retina from Cav1.4$^{-/-}$ mice. Cav1.4 expression in human T cells was difficult to assess because of a very strong band detected by the Cav1.4 antibody in human T cells that ran just below the Cav1.4 reference band in retina along with some weaker bands above the Cav1.4 reference band.

Given the limited specificity of most antibodies for all but some VGCC α$_1$ subunits, we also investigated their expression by analyzing RNA-Seq data of human and mouse T cells from a HD or WT mice, respectively, that were left untreated or stimulated by TCR crosslinking. We focused on VGCCs whose TPM normalized mRNA expression values exceeded 1 in T cells. These included, in descending order of total mRNA expression, mouse *Cacna1a* (Cav2.1), *Cacna1i* (Cav3.3) and *Cacna1h* (Cav3.2), as well as human *CACNA1I* (Cav3.3), *CACNA1H* (Cav3.2) and *CACNA1F* (Cav1.4) (Fig. 7A, B). Except for human Cav3.3, mRNA levels of VGCCs in T cells were markedly lower than those in reference tissues known to express VGCCs (brain, heart, skeletal muscle, retina). A similar mRNA expression pattern of α$_1$ subunits was observed in mouse CD4$^+$ T cells that had been polarized in vitro into Th1, Th2, Th17 or iTreg cells as well as naturally occurring Treg cells[47] (Supplementary Fig. 10A). Again, *Cacna1a* (Cav2.1) was the most highly expressed α$_1$ subunit across all CD4$^+$ T cell subsets. Compared to other T cell subsets, relatively higher mRNA levels were observed for *Cacna1c* (Cav1.2) in Th2 cells and *Cacna1g* (Cav3.1) in Treg cells (Supplementary Fig. 10B). Because the α1 channel subunits Cav1.2, Cav1.4 and Cav3.1 have been implicated in the function of Th2 and Th17 cells, respectively[15,19,46,48], we tested whether the differentiation of naïve CD4$^+$ T cells into these T cell subsets is associated with the occurrence of voltage activated Ca$^{2+}$ influx. The polarization of mouse CD4$^+$ T cells into Th2, Th17 and induced Treg (iTreg) cells was associated with the expected upregulation of lineage-specific transcription factors including GATA3 (Th2), Foxp3 (iTreg) and RORγt (Th17) (Supplementary Fig. 10C, D). Depolarization of Th2, Th17 and iTreg cells with 60 mM or 150 mM K$^+$ in the extracellular buffer, however, failed to evoke detectable Ca$^{2+}$ influx (Supplementary Fig. 10E, F). By contrast, ionomycin induced robust SOCE in all T cell subsets, which was suppressed by high extracellular K$^+$ as expected.

Given the apparent expression of several α$_1$ subunits of VGCCs, we hypothesized that T cells may express alternatively spliced and non-functional isoforms of VGCCs. Indeed, splice variants of Cav1.1 and Cav1.4 have been reported in T cells[49,50]. We conducted an exon-level alignment of RNA-seq data for those VGCCs that we had found to be expressed in human and mouse T cells. mRNA for Cav3.3 is the most abundant of all VGCCs in human T cells. However, only transcripts of exons 12-37 were detectable in human T cells, which was in contrast to brain (frontal cortex) where all 37 exons of Cav3.3 (encoded by *CACNA1I*) are expressed (Fig. 7C). We found two putative transcription start sites (TSS) 5′ of exon 12 in human T cells by searching the refTSS dataset[51] (Fig. 7D). We made similar observations for Cav3.2 (*CACNA1H*), which is the second highest expressed VGCC in human T cells. Transcript levels of exons 1-12 were undetectable or very low in human T cells, whereas all exons (1-35) were expressed in brain tissue (Fig. 7E). We detected a putative TSS in exon 13, which may initiate mRNA expression in T cells (Fig. 7F). Cav2.1 (encoded by *Cacna1a*) is the most highly expressed α$_1$ subunit in mouse T cells at the transcriptional level. Exon usage analysis demonstrated that exons 1-33 (of 49) were not or weakly expressed in T cells (Fig. 7G). We detected 3 putative TSS in exon 34 of mouse *Cacna1a*, which may initiate transcription of a truncated mRNA (Fig. 7H). Collectively, these data demonstrate that although mRNAs for several VGCCs can be detected in mouse and human T cells, the transcripts are incomplete and result in N-terminally truncated proteins. For example, the non-transcribed exons 1-11 of *CACNA1I* encode amino acids (aa) 1-715 of the Cav3.3 protein, which form the N terminus of Cav3.3, its first channel domain (I), the I–II linker including the α-interaction domain (AID) and TM1-3 of the second channel domain (II) (Fig. 7H). Our data predict similar N-terminally truncated proteins for human Cav3.2 and mouse Cav2.1. Even if these proteins were stable and properly located in the plasma membrane, they would very likely not be functional Ca$^{2+}$ channels, providing an explanation for the absence of VGCC currents and Ca$^{2+}$ influx upon depolarization in T cells.

## Discussion

We here identified Cavβ1 as a regulator of clonal expansion of T cells. By using an shRNA screening approach to identify ion channels that control T cell-mediated immunity, we found that Cavβ1 was required for the clonal expansion of CD4$^+$ T cells after LCMV infection in vivo. Whereas deletion of Cavβ1 did not affect the proliferation of T cells, it was required to prevent T cell apoptosis following TCR stimulation in vitro. Three other Cavβ

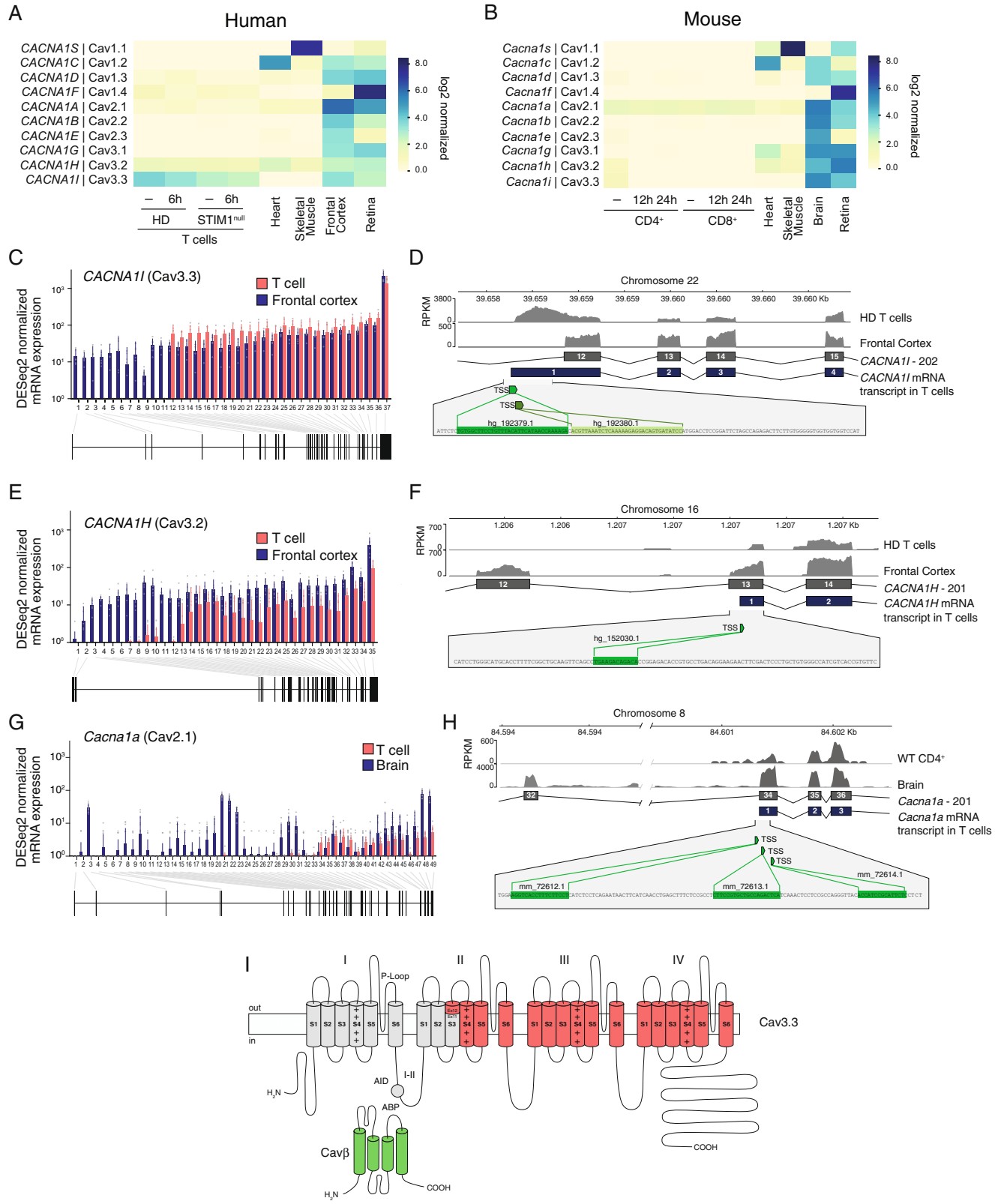

subunits have also been implicated in T cell function. Lack of functional Cavβ2 and Cavβ4 was associated with a severe defect in T cell development[20], thymic and splenic involution and lymphocytopenia[22]. Cavβ3 deficiency resulted in impaired survival of naive CD8[+] T cells[24]. Together, these studies and our data indicate that Cavβ subunits regulate T cell function, especially T cell survival. In contrast to the Cavβ1 deficiency

phenotype described here, deletion or mutation of Cavβ2, Cavβ3 or Cavβ4 subunits resulted in moderately reduced TCR-induced Ca[2+] influx in T cells[20,23,24], which was explained as arising from reduced Cav1.2/Cav1.3 and Cav1.4 protein expression in Cavβ2 and Cavβ3 deficient T cells, respectively[20,24]. It is noteworthy that we only observed robust expression of Cavβ1 and Cavβ3 in mouse and human T cells by RNA-Seq, but not that of Cavβ2 and

**Fig. 7 Lack of full-length transcripts of VGCC α₁ subunits in T cells. A, B** Absolute mRNA expression levels of α₁ subunits of VGCCs in human (**A**) and mouse (**B**) T cells and reference tissues. **A** CD4⁺ T cells were left unstimulated (−) or stimulated for 6 h with anti-CD3 + CD28 antibodies. HD represents the averaged data from three individual healthy donors (HD) and a patient with a *STIM1* null mutation (c.497 + 776 A > G; STIM1^null). **B** Mouse CD4⁺ and CD8⁺ T cells were left unstimulated (−) or stimulated for 12 or 24 h with anti-CD3 + CD28 antibodies. VGCC expression in T cells in (**A, B**) was analyzed by RNA sequencing and compared to that in human and mouse heart, skeletal muscle, frontal cortex, whole brain and retina using published datasets (Supplementary Table 2). **C–F** Exon usage of human *CACNA1I* (Cav3.3) (in **C, D**) and *CACNA1H* (Cav3.2) (in **E, F**) in CD4⁺ T cells and frontal cortex. **C, E** Normalized mRNA expression in unstimulated T cells from n = 3 HDs (averaged; red) and frontal cortex (blue) per exon. **D, F** Transcript levels (as RPKM, reads per kb of transcript, per million mapped reads) in frontal cortex and CD4⁺ T cells from an individual HD superimposed on exons 12–15 (for CACNA1I in **D**) and exons 12–14 (for CACNA1H in **F**). In T cells, exon 12 and exon 13 are the first transcribed exons of CACNA1I and CACNA1H, respectively. Green boxes and nucleotide sequences indicate predicted alternative transcriptional start sites (TSS). **G** Exon usage of *Cacna1a* (Cav2.1) in mouse CD4⁺ T cells (red) and brain (blue). Normalized *Cacna1a* mRNA expression per exon and exon-intron structure. **H** *Cacna1a* transcript levels (as RPKM) in brain and CD4⁺ T cells from mice superimposed on exons 32–36. In T cells, exon 34 is the first transcribed exon. Green boxes and nucleotide sequences indicate predicted alternative TSS. **I** Membrane topology model of the predicted N-terminally truncated Cav3.3 protein in T cells lacking channel domain I and part of domain II. AID α₁ interacting domain, ABP α₁ binding pocket.

Cavβ4. Because we analyzed T cells from secondary lymphoid organs and blood of mice and humans, respectively, it is possible that Cavβ1 and Cavβ3 regulate the function of mature T cells such as survival, whereas Cavβ2 and Cavβ4 are critical in immature T cells during their development.

The mechanisms by which Cavβ1 controls apoptosis remain unclear. In excitable cells Cavβ subunits regulate VGCC function by promoting the cell surface expression of α₁ subunits and controlling channel activation and inactivation[7]. Reduced surface expression or activation of α1 subunits in the absence of Cavβ1 might therefore result in impaired Ca²⁺ influx in T cells and explain impaired T cell survival. However, Cavβ1 deletion in T cells did not impair TCR-induced Ca²⁺ influx, suggesting that Cavβ1 function is independent of regulating VGCC channels. Ca²⁺ signaling is an important regulator of apoptosis in T cells with both pro- and anti-apoptotic effects observed[52]. However, our observation that Ca²⁺ influx in T cells is unaffected by the loss of Cavβ1 excludes the possibility that increased apoptosis is caused by effects of Cavβ1 on Ca²⁺ signals. Cavβ1-deficient T cells also showed normal production of cytokines such as IFN-γ, TNF and IL-2 whose transcription is dependent on Ca²⁺ signaling[2,53], further demonstrating that Cavβ1 is not required for Ca²⁺ influx in mouse T cells.

Although Cavβ proteins have mostly been thought of in terms of auxiliary subunits of VGCCs, a significant body of evidence demonstrates that they interact with many other proteins and have many VGCC-independent functions[7,26]. For instance, Cavβ proteins interact with other ion channels including ryanodine receptors, membrane receptors, Ras-related monomeric small GTP-binding (RGK) proteins, dynamin, actin and the scaffolding protein AHNAK1[26]. One of the most intriguing functions of Cavβ proteins is their function in controlling gene expression in the nucleus. A Cavβ4 splice variant was shown to interact with heterochromatin protein 1 γ (HP1γ), which mediates gene silencing[54]. Full-length Cavβ3 interacts with Pax6(S), an isoform of the transcription factor Pax6, to repress its transcriptional activity[55]. Moreover, overexpression of Cavβ4 in HEK293 cells was shown to modulate gene expression[56]. The Cavβ1a isoform was shown to localize to the nucleus of muscle progenitor cells (MPC) and bind to the myogenin promoter[57]. Deletion of Cavβ1a altered MPC expansion in vitro and in vivo, and changed global gene expression[57]. It is possible that Cavβ1 also controls gene expression in T cells, which would be distinct from its canonical purpose of regulating VGCC function in excitable cells[58].

Because we identified Cavβ1 as a regulator of T cell function and previous reports had implicated other α₁ and β subunits in Ca²⁺ influx and the function of T cells, we investigated the contribution of VGCCs to Ca²⁺ signaling in T cells. Evidence supporting a function of VGCCs in T cells comes from the use of

DHP Ca²⁺ channel blockers, RNAi mediated knockdown of VGCC expression and knockout mice[15,17,19,20,23,24,59–61]. Using a variety of measurement protocols in both human and mouse T cells, we were unable, however, to detect any evidence of functional VGCCs in T cells. Depolarization of T cells failed to evoke Ca²⁺ influx and VGCC currents even under electrophysiological recording conditions greatly optimized to detect small currents. Similar results were obtained in T cells simultaneously activated by TCR stimulation based on the hypothesis that depolarization may not be sufficient to activate VGCCs in T cells and that additional stimuli may be required for their activation. Moreover, VGCC currents were undetectable in T cells lacking CRAC channel function. Because CRAC channels are the dominant Ca²⁺ influx pathway in T cells after TCR stimulation, we reasoned that lack of CRAC channel function may result in a compensatory upregulation of VGCC currents, which was not the case. Lastly, deletion of STIM1, which was reported to inhibit Cav1.2 ectopically expressed in HEK293 or Jurkat cells[44,45], did not evoke Ca²⁺ influx upon depolarization of mouse or human T cells. Together, our studies fail to provide evidence for the existence of functional VGCCs in T cells.

How can these findings be reconciled with reports of VGCC function in T cells? Initial evidence supporting VGCC function in T cells came from the use of VGCC blockers including DHPs (amlodipine, nicardipine, nimodipine) and non-DHPs (verapamil, diltiazem), which were reported to inhibit Ca²⁺ influx in T cells[60–63]. Micromolar concentrations of VGCC blockers, however, also inhibit several K⁺ channels, notably Kv1.3 in T cells, thereby reducing Vₘ and the electrical driving force for Ca²⁺ influx through CRAC channels[35,64]. In fact, selective blockade of Kv1.3 and KCa3.1 channels in T cells inhibits Ca²⁺ influx[65–67]. It is noteworthy that there is no compelling evidence showing that patients treated with VGCC blockers have impaired immune responses despite the widespread clinical use of these drugs for the treatment of many cardiovascular conditions[68]. More specific evidence supporting VGCC function in immunity comes from studies in mice. Targeted disruption of *Cacna1f* encoding Cav1.4 resulted in reduced CD4⁺ and CD8⁺ T cell numbers and function, most strikingly impaired T cell responses after infection with *L. monocytogenes* and murine gammaherpesvirus 68 (MHV-68)[17,18]. T cells of *Cacna1f*⁻/⁻ mice had reduced TCR stimulation-induced Ca²⁺ influx and voltage-dependent Ba²⁺ currents[17]. We were not able to observe Ba²⁺ currents upon depolarization of human peripheral blood T cells. Although this discrepancy could be due to a specific function of Cav1.4 in mouse but not human T cells, we failed to observe depolarization-induced Ca²⁺ influx in both human and mouse T cells. Another study showed that deletion of *Cacna1g* encoding Cav3.1 impairs the production of cytokines by Th17 cells in vitro

and renders mice resistant to EAE, a mouse model of multiple sclerosis[19]. CD4+ T cells of Cacna1g−/− mice had reduced voltage-activated Ca2+ currents and Ca2+ influx when exposed to extracellular Ca2+. This defect was more prominent in Th17 cells, which expressed higher levels of Cacna1g mRNA than Th1 or Th2 cells. By contrast, no defect in Ca2+ influx was observed in Cacna1g−/− T cells after TCR stimulation. This study suggested that Cav3.1 channels mediate Ca2+ influx at the resting membrane potential in T cells, especially in Th17 cells.

These findings regarding Cav1.4 and Cav3.1 functions in mouse T cells[17,19] are in apparent contrast to our failure to detect VGCC currents in human peripheral blood T cells. A potential explanation for this discrepancy could be that mouse T cells have functional VGCCs whereas human T cells do not. Arguing against this explanation is the fact that we failed to detect voltage-dependent Ca2+ influx in mouse T cells under any of the conditions tested. To activate VGCCs, we depolarized human and mouse T cells to ∼ −24 mV and ∼0 mV (with 60 mM and 150 mM [K+]o, respectively), which would be sufficient to activate L-type (Cav1.4) and T-type (Cav3.1) channels that open at membrane potentials between −40 to −10 mV and −60 to −70 mV, respectively. Depolarization should, therefore, have induced voltage-dependent Ca2+ currents and Ca2+ influx in our experiments. An intriguing feature of T-type VGCCs such Cav3.1 that may explain their function in T cells is their activation at low voltages in the −60 to −70 mV range, which coincides with the resting $V_m$ of T cells (−53 to −59 mV)[69]. Window currents at these membrane potentials result from the overlap of activation and inactivation of T-type channels, and in the CNS were found to be important for the regulation of neuronal arousal[70,71]. In T cells, they may contribute to Ca2+ influx and T cell function at resting $V_m$ and explain the effects of Cav3.1 deletion on T cell function in Cacna1g−/− mice. Notwithstanding the potential importance of window currents for T cell function, if functional T-type VGCCs were expressed in T cells, we should have been able to detect Ca2+ currents and Ca2+ influx upon depolarization. Our inability to detect T-type VGCC currents in mouse T cells is consistent with the lack of Cav3.1 mRNA expression in mouse or human T cells.

Our RNA-Seq data demonstrated transcription of several genes encoding α1 pore subunits in human and mouse T cells, but the α1 pore subunits we detected were different from those reported before, namely Cav1.2, Cav1.3, Cav1.4 and Cav3.1[15,17,19,46]. The most abundant α1 subunits we detected were Cav3.3 and Cav2.1 in human and mouse T cells, respectively, whereas transcripts encoding Cav1.2, Cav1.3, Cav1.4 and Cav3.1 were not or barely detectable. Moreover, we were unable to detect Cav1.2 or Cav1.4 proteins in either mouse or human T cells. The analysis of protein expression of other α1 subunits was limited by the quality of available antibodies, which included the validation of Cav3.3 protein expression. A potential explanation for our inability to detect VGCC function and mRNA of previously reported Cav α1 subunits is that their expression is restricted to particular T cell subsets. Cav1.2 levels were reported to be increased in human Th2 cells compared to Th1 or Th0 cells[46], and Cav3.1 mRNA levels were transiently increased in Th17 cells[19]. Our analysis of published RNA-Seq data of mouse CD4+ T cells[47] confirms relatively higher mRNA levels for Cacna1c (Cav1.2) and Cacna1g (Cav3.1) in Th2 and Treg cells, respectively, compared to naïve T cells, Th1 and Th17 cells. However, electrophysiological evidence of functional VGCCs in Th2 or Treg cells is currently lacking.

Intriguingly, neither Cav3.3 and Cav2.1 have been implicated in T cell function before. To understand why their robust mRNA expression was not associated with VGCC function, we took advantage of the fact that RNA-sequencing reads can be aligned to individual exons to determine exon usage and potential splice variants in T cells. Whereas mRNA corresponding to all 37 exons

of human CACNA1I was detected in the brain, where Cav3.3 function has been reported[72,73], only mRNA corresponding to exons 12–37 of CACNA1I was detectable in human T cells. We observed a similar lack of transcription of 5′ exons for human CACNA1H (Cav3.2) and mouse Cacna1a (Cav2.1). We detected two TSS preceding exon 12 of CACNA1I and a TSS in exon 13 of CACNA1H, which coincide with the 5′ end of the mRNA transcripts in T cells. These 5′ truncated mRNAs are predicted to encode proteins that lack the NH2 terminus of the α1 subunits they encode including the entire channel domain I, the cytosolic I–II linker and part of domain II. It is unlikely, however, that the truncated proteins are stable or properly localize in the plasma membrane. While it is theoretically conceivable that a protein comprising domains III and IV of Cav3.3 is expressed and may assemble into a homomeric or heteromeric Ca2+ channel that is gated by a voltage-independent mechanism, we consider this possibility to be remote.

Alternative splicing of VGCC α1 subunits is well documented and an important mechanism to produce channels with distinct functional properties[5]. However, alternative splicing typically occurs in the C-terminus, and involves the alternative usage of individual exons. For instance, alternative splicing of Cav3.3 was shown to involve exon 9 encoding the I–II linker domain and exons 33 and 34 encoding part of the C-terminus of the channel, thereby giving rise to variants with distinct biophysical properties[74,75]. Alternative splicing of VGCCs has also been reported in T cells[49,50]. These include two variants of Cav1.4 in human T cells that lack expression of exons 31–34 & 37, and 32 & 37, respectively, predicted to delete the VSD of domain IV and its DHP binding site, which may render the channel insensitive to depolarization[49]. The alternative exon usage of human CACNA1I and CACNA1H as well as mouse Cacna1f in T cells we demonstrate here has not been reported before. Of note, Man et al. recently reported that human and mouse cardiomyocytes do not express full-length transcripts of the neuronal voltage-gated sodium channel Nav1.8 (Scn10a), but instead express a short transcript (Scn10a-short) comprising only the last 7 exons[76]. Transcription of this short variant occurred from an intronic enhancer-promoter complex. Overexpression of Nav1.8-short protein was shown to modulate the function of the main cardiac sodium channel Nav1.5 and heart rhythm. Whether truncated Cav3.3 protein is expressed in human T cells and suppresses VGCC function remains to be elucidated.

While our study does not support the existence of functional VGCCs in T cells, it suggests that Cavβ1 has alternative, VGCC-independent functions in T cells. These findings have implications for the assessment of the safety of VGCC channel blockers that are in wide clinical use for cardiovascular diseases, which based on our data are not expected to result in suppression of immune function. This interpretation is consistent with a lack of clinical evidence for immunosuppression in patients treated with Ca2+ channel blockers.

## Methods

**Mice.** All experiments were conducted in accordance with protocols approved by the Institutional Animal Care and Use Committee of at New York University Grossman School of Medicine. Stim1fl/fl Cd4Cre[77], Cacna1f-mutant[78], congenic CD45.1+ SMARTA[79] have been described previously. Congenic CD45.2+ (strain 000664), Cd4Cre (strain 017336) and Rosa26-LSL-Cas9 knock-in (strain 024857) mice were purchased from the Jackson laboratory (Bar Harbor, ME). SMARTA; LSL-Cas9; Cd4Cre mice were generated by crossing SMARTA, Cd4Cre and LSL-Cas9 mice. All animals were on a pure C57BL/6 genetic background. Male and Female mice were used between 8 and 16 weeks of age. Mice were maintained under specific pathogen-free conditions with a 12 h dark/light cycle, at 22–25 °C and 50–60% humidity with water and food provided ad libitum.

**Human cells.** Experiments using human cells were conducted in accordance with protocols approved by the Institutional Review Board of the New York University

Grossman School of Medicine. Informed consent for the studies was obtained from the patients' parents and HDs in accordance with the Declaration of Helsinki. T cells from HDs and a patient homozygous for an autosomal recessive *ORAI1* p.R91W LOF mutation (referred to as ORAI1[LOF]) were isolated from whole blood and stimulated as described[42]. Peripheral blood mononuclear cells were isolated from a 7yr-old male patient and his father, who were homozygous and heterozygous, respectively, for a cryptic splice acceptor site mutation in an intronic region between exons 4 and 5 of *STIM1* (c.497 + 776 A > G). The mutation (referred to as STIM1[null]) abolished STIM1 protein expression and store-operated $Ca^{2+}$ influx (SOCE), and caused CRAC channelopathy syndrome in the patient[80]. A detailed case report describing the phenotype and molecular characterization of this patient will be published in a forthcoming manuscript.

**Cell lines**. HEK293 cells were cultured in full DMEM medium supplemented with 10% FBS (Sigma-Aldrich, 12306 C) and 1% penicillin plus streptomycin (Gibco, 15140-122). Platinum E (Plat E) packaging cells were cultured in full DMEM medium supplemented with 10 µg/ml Blasticidin S (Invivogen, ant-bl-1) and 1 µg/ml Puromycin (Gibco, A11138-03). The PC12 cell line was cultured at 37 °C, 5% $CO_2$ and passaged twice a week, and the experiments were performed on cells at passage 17 for electrophysiology experiments. Cells were maintained in DMEM containing 10% fetal calf serum (HyClone), 1% penicillin/streptomycin. After removing the supernatant, PC12 cells were washed with PBS, then treated with 0.05% Trypsin-EDTA (Life Technologies, 25300054) and incubated for 5 min at 37 °C. Next, the cells were transferred to a centrifuge tube, complete medium was added (to inactivate Trypsin), and cells were centrifuged for 5 min at $250 \times g$. The supernatant was removed, and cells were resuspended in fresh medium. Cells were plated onto poly-D-lysine coated coverslips for patch clamp experiments. HEK293 cells were maintained in suspension at 37 °C with 5% $CO_2$ in CD293 medium (Thermo Fisher Scientific, 11913019) supplemented with 4 mM GlutaMAX (Thermo Fisher Scientific, 35050-061). For electrophysiology, cells were plated onto poly-L-lysine coated coverslips one day before transfection and grown in a medium containing DMEM/F12 (Corning: 0-090-CV), 10% fetal bovine serum (Corning, 35-011-CV), 2 mM glutamine, 50 U/ml penicillin and 50 µg/ml streptomycin.

**Transfections and production of pseudotyped retrovirus**. HEK293 cells were transfected with rCAV1.2 WT (150 ng), pcDNA3.1 α2δ1 (75 ng), pcDNA3.1 β2 α (75 ng) and Cherry-C1 (20 ng) kindly provided by Dr. R.W. Tsien (NYU) using Lipofectamine 2000 (Thermo Fisher Scientific). Cells were used for electrophysiology 24–48 h after transfection. To produce pseudotyped retrovirus for transduction of T cells, Plat E cells were transfected with retroviral expression plasmids encoding shRNAs (pLMPd-Amt, pLMPd-GFP)[81] and small guide RNAs for CRISPR/Cas9 gene editing (pMIR-Amt, pMRI-GFP)[82]. Plat E cells were cotransfected with the ecotropic packaging vector pCL-Eco using GenJet lipofection reagent (SignaGen, SL100489). Retroviral supernatant was collected 36 and 60 h after transfection.

**T cell culture**. Human T cells were separated from whole blood by density gradient centrifugation using Ficoll-Paque plus (GE Amersham) and expanded in vitro as previously described[83]. Mouse CD4[+] T cells were purified from splenocytes using the MagniSort Mouse CD4[+] T Cell Enrichment Kit (Invitrogen, MS22-7762-74) according to manufacturer's protocol. CD4[+] T cells were stimulated in flat bottom 12-well plates ($1 \times 10^6$ cells/ml per well) with 1 µg/ml plate-bound anti-CD3 (Bio X cell, clone 2C11, 14-0031-85) and 1 µg/ml anti-CD28 antibodies (Bio X cell, clone 37.5, BE0015-1) in complete RPMI medium (Corning, 10-040-CV) containing 10% FBS, 1% L-glutamine, 1% penicillin-streptomycin and 0.1% β-mercaptoethanol. After 3 days of stimulation, T cells were detached and transferred to a new plate with fresh complete RPMI medium containing 20 IU/ml rh-IL-2 (Peprotech, 200-02) and 2.5 ng/ml IL-7 (Peprotech, 217-17). For differentiation of naive CD4[+] T cells into Th0, Th2, Th17 or iTreg cells, T cells stimulated with plate-bound anti-CD3 + CD28 and cultured for 3-5 days in the presence of 5 µg/ml anti-IFN-γ (clone XMG1.2) and 5 µg/ml anti-IL-4 (clone 11B11; both eBioscience) for Th0; 100 ng/ml IL-4 (PeproTech), 5 µg/ml anti-IFN-γ, and 5 µg/ml anti-IL-12 (eBioscience) for Th2 cells; 20 ng/ml IL-6 (R&D Systems), 0.5 ng/ml hTGF-β (PeproTech), 5 µg/ml anti-IFN-γ, and 5 µg/ml anti-IL-4 for Th17 cells; 50 IU/ml IL-2 (NIH), 2 ng/ml hTGF-β, 5 µg/ml anti-IFN-γ, and 5 µg/ml anti-IL-4 for iTreg cells. For retroviral transduction, mouse T cells were transduced 24 h after stimulation with anti-CD3/anti-CD28 antibodies by spin-infection ($1.450 \times g$, 90 min, 32 °C) in the presence of retroviral supernatant and 8 µg/ml polybrene (EMD Millipore, TR-1003-G). Retroviral supernatant from T cells was diluted 1:2 with complete RPMI containing 20 IU/ml rh-IL-2 and 2.5 ng/ml IL-7 30 min after spin infection, and replaced with fresh complete RPMI containing IL-2 and IL-7 16 h later. Transduced T cells were analyzed by flow cytometry using Ametrine or GFP reporters 3 days after spin infection.

**Generation of shRNA library targeting ICTs**. The shRNA library targeting 223 ion channels, transporters and regulatory proteins (ICTs) was custom-generated based on the analysis of mRNA expression levels of 658 ICTs in human and mouse immune cells using the Immunological Genome Project (ImmGen)[27] and Fantom5[28] databases. 223 ICTs were determined to be expressed at least twofold

above the population average across all immune cell types and included in the shRNA library. Each ICT was targeted by five shRNAs. Also included in the pooled shRNA library were 34 positive controls (genes known to regulate T cell expansion and survival) and 13 negative controls (genes not expressed in mammalian cells nor in T cells). shRNAs were designed as described in[84]. Briefly, siRNAs targeting 223 ICTs were designed using the DSIR algorithm[85] and filtered to select shRNAs with effective shRNAmir processing and potent knockdown[86]. For de novo generation of shRNA plasmids, 97-mer oligonucleotides (IDT Ultramers) coding for the respective shRNAs were synthesized on 55k arrays (Agilent) and cloned into the pLMPd recipient vector, which is based on miR-E vector and encodes Ametrine (Amt) as fluorescent reporter[81]. The pooled shRNA library containing 1342 shRNAs was sequenced by HiSeq 2500 (Illumina) to confirm equal representation of shRNAs and subsequently used to transfect Plat E cells for the production of pseudotyped retroviruses.

**In vivo shRNA screen**. LCMV-specific CD4[+] T cells isolated from the spleens of CD45.1[+] SMARTA mice were purified using the MagniSort Mouse CD4[+] T cell Enrichment Kit and stimulated with plate-bound anti-CD3 and anti-CD28 antibodies. T cells were transduced 24 h after stimulation with the shRNA library packaged in pseudotyped retroviral particles at ~0.3 multiplicity of infection in the presence of 8 µg/ml polybrene. 30 min after spin infection, the retroviral supernatant was diluted 1:2 with complete RPMI containing 20 IU/ml rh-IL-2 and 2.5 ng/ml IL-7. 16 h later, the T cell supernatant was replaced with fresh complete RPMI containing IL-2 and IL-7 16 h later. 3 days after transduction, transduced T cells (Amt[+]) were enriched by cell sorting using a SY3200 (HAPS1) cell sorter (Sony) under sterile conditions. $1 \times 10^6$ Amt[+] SMARTA T cells were retro-orbitally injected into congenic CD45.2[+] host mice. $1 \times 10^6$ Amt[+] SMARTA T cells were kept as input control. 2 days after adoptive T cell transfer, host mice were infected i.p. with $2 \times 10^5$ PFU of LCMV Armstrong (LCMV[ARM]). 7 days later, CD45.1[+] donor T cells were isolated from the spleen of host mice and CD4[+] CD45.1[+] Amt[+] T cells were purified by cell sorting. Genomic DNA (gDNA) was isolated from sorted cells using the FlexiGene DNA Kit (Qiagen, 51204) following the manufacturer's instructions. Next generation sequencing (NGS) libraries were generated from gDNA by PCR amplification of shRNA guide strands using primers that tag the product with Illumina HiSeq 2500 adaptors (*primerA* + mirE, *AATGA-TACGGCGACCACCGA*GAATTCTAGCCCCTTGAAGTC; *primerB* + BarCode + mirE-Loop, *CAAGCAGAAGACGGCATACGA*XXXXXTAGTGAAGCCACA-GATGTA). shRNAs were amplified in at least three individual 50 µl PCR reactions containing 500 ng of gDNA, 1U Platinum *pfx* DNA polymerase (Thermo Scientific, 11708013), 1× Platinum *pfx* buffer, 0.3 µM primers and 0.3 mM dNTPs using the following conditions: 94 °C for 5 min; 35 cycles of 94 °C for 15 s, 55 °C for 30 s and 68 °C for 20 s; 72 °C for 7 min. The amplified shRNA library (~122 bp) was purified by phenol extraction and using the QIAEX II Gel Extraction Kit (Qiagen, 20021). The quality of the amplified library was assessed using the Agilent 2200 TapeStation Bioanalyzer (Agilent Technologies). The libraries were sequenced using the custom mirE-EcoR1 sequencing primer TAGCCCCTTGAAGTCCGAGGCAG-TAGGCA and the single read 50 bp (SR50) rapid-mode of the HiSeq 2500 DNA Sequencer (Illumina). Sequencing results were analyzed using the Model-based Analysis of Genome-wide CRISPR-Cas9 Knockout (MAGeCK) package[87] to identify shRNAs that were depleted or enriched in donor T cells after LCMV infection compared to input T cell samples.

**Design and cloning of shRNAs and sgRNAs**. shRNA target sequences against mouse *Cacnb1* were extracted from the shRNA library. sgRNA target sequences were designed using Benchling software (https://www.benchling.com/crispr/). In addition, shRNA or sgRNA targeting a non-mammalian gene (Renilla luciferase) or human VEGF were used as controls. shRNA and sgRNA were cloned into the pLMPd[81] and pMRI[82] retroviral expression vectors, respectively, that encode Ametrine or GFP reporters. shRNA and shRNA sequences are provided in Supplementary Table 1.

**RNA-sequencing**. RNA-Seq data derived from human and mouse tissues were extracted from the Gene Expression Omnibus (GEO) database and are listed in Supplementary Table 2. FASTQ files were trimmed with Trimmomatic v0.36[88] and aligned with STAR/2.6.1[89] to the corresponding human GRCh38/hg38 or mouse GRCm38/mm10 genome assembly. BigWig files were generated using deeptools v3.1.0[90] and normalized using Reads Per Kilobase of transcript per Million mapped (RPKM). Human transcripts were counted using Salmon v0.14.1[91] with annotation from Gencode v30 Gene Transfer Format (GTF) file, and mouse transcripts were aligned and counted using the featureCounts function in the subread package v1.6.3[92] with annotation from Gencode Gene Transfer M21 (GTF GRCm38.p6) format file. The final data to generate heatmaps of gene expression for human and mouse samples was TPM normalized to allow comparison of expression levels between different genes. The final heatmap visualization was done in python using the mwaskom/seaborn: v0.8.1 package[93].

Additional RNA-Seq data were generated using human CD4[+] T cells from a STIM1[null] patient (homozygous for the *STIM1* c.497 + 776 A > G mutation) as well as his healthy brother and father (both heterozygous for the *STIM1* mutation) and an unrelated healthy donor. CD4[+] T cells were left untreated or stimulated with

5 µg/ml plate-bound anti-CD3 (clone OKT3, 14-0037-82) and 10 µg/ml soluble anti-CD28 (clone CD28.2, 14-0289-82, both eBioscience) antibodies for 6 h in RPMI1640 medium supplemented with 10% FBS, 1% L-glutamine, and 1% penicillin-streptomycin. Total RNA was extracted using the RNeasy Micro Kit (Qiagen, 74004) and RNA quality was analyzed using a Bioanalyzer 2100 (Agilent) using PICO chips. RNA-Seq libraries were prepared using the TruSeq RNA sample prep v2 kit (Illumina TruSeq Stranded mRNA, RS-122-2001) and 100 ng total RNA following the manufacturer's instructions. The amplified libraries were purified using AMPure beads (Beckman Coulter, A63881), quantified by Qubit 2.0 fluorometer (Life Technologies), and visualized using an Agilent Tapestation 2200. The libraries were pooled equimolarly, loaded on the HiSeq 2500 DNA Sequencer and run as single 50 nucleotide reads. For analysis of VGCC expression, data from the healthy donor, brother and father of the STIM1[null] patient were averaged and are referred to as HD (healthy donor).

**ICT expression and correlation analysis**. ICT mRNA expression in human CD4[+] and CD8[+] T cells was analyzed using publicly available RNA-Seq datasets GSE87508[94] and GSE133822[95] as well as data from Haemopedia (https://www.haemosphere.org/)[96]. Raw data from Haemopedia and GSE133822 were normalized using the transcripts per million (TPM) method, and GSE87508 data was normalized using the Relative Log Expression method. Heatmaps, scatterplots and statistics were generated using MATLAB (R2019, Mathworks).

**Exon usage analysis and transcriptional start site analysis**. The exon-level quantification of VGCC mRNA expression in human and mouse tissues and cells was performed using the HTSeq package (v0.11.2)[97] with annotations from the human (GRCh38.p12) Gencode v30 GTF file and mouse (GRCm38.p6) Gencode M25 GTF file, respectively. The exon-level RNA expression data was normalized using the median of ratios method from DESeq2[98] Bioconductor package in R (version 3.6.1). This method was used to allow comparison between RNA expression per exon in T cells and reference tissues. Bars graphs showing quantification of DESeq2 normalized mRNA expression for each exon were generated using MATLAB R2019. The visualization of bigWig files generated from aligned reads after performing RPKM normalization was done using pyGenomeTracks[99]. Potential TSS near the 5′ end of mRNA transcripts in T cells were obtained from refTSS[51].

**Flow cytometry**. Cells from tissue culture or isolated from mouse spleens were washed in cold PBS containing 3% FBS and 2 mM EDTA ("FACS buffer"). Staining of cell surface molecules with fluorescently labelled antibodies was performed at RT for 10 min in the dark. Intracellular (IC) cytokine staining was performed using the IC staining buffer kit (Invitrogen, 88-8824-00) according to the manufacturer's instructions. All antibodies were used at 1:200 dilution and the complete list of antibodies used in this study can be found in Supplementary Table 3. Briefly, T cells transduced with sgRNAs or shRNAs were stimulated with 1 µM ionomycin (Invitrogen, I-24222) and 20 nM phorbol myristate acetate (PMA, Calbiochem, 524400) for 6 h in the presence of brefeldin A (eBioscience, 00-4506-51). For apoptosis measurements, cells were incubated for 10 min at RT with Annexin V-AF647 in Annexin V Binding buffer (BioLegend, 422201). Samples were acquired using FACSDiva on a LSR Fortessa cell analyzer (BD Biosciences) and analyzed using FlowJo software (FlowJo 10.5.3). Cell sorting of transduced, Ametrine[+] T cells was performed using a MoFlo XDP cell sorter (Beckman Coulter).

**Intracellular Ca$^{2+}$ measurements**. T cells were loaded with the Ca$^{2+}$ sensitive dye Fura-2 (Invitrogen, F1221) for 30 min in complete RPMI1640 medium, and analyzed either using single-cell time-lapse microscopy or a microplate reader. For time-lapse microscopy, human T cells were loaded with 1 µM Fura-2-AM for 30 min and co-stained with anti-CD4-PE (Biolegend, 300550) and anti-CD8-FITC (Biolegend, 344704) at 1:200 dilution in the last 10 min of incubation, plated onto UV-sterilized coverslips pre-coated with 0.01% poly-L-lysine (Sigma-Aldrich, P8920) and perfused with buffer solutions in a RC-20 flow chamber (Warner Instruments). The chamber was mounted on an IX81 inverted epifluorescence microscope (Olympus). Some time-lapse imaging experiments were performed simultaneously with patch-clamping electrophysiology using a Nikon Diaphot TMD microscope equipped with a Nikon Fluor objective (NA 1.3). Fura-2 emission at 510 nm was measured following excitation at 340 nm and 380 nm using a Lambda DG-4 Plus light source and wavelength switcher (Sutter Instruments). F340/F380 emission ratios were calculated for each time point, and region of interests were used to discriminate human CD4 and CD8 T cells using the red (RFP) and green (GFP) filters, respectively, using SlideBook imaging software v4.2 (Olympus). Baseline Ca$^{2+}$ levels were determined after washing cells with 2 mM Ca$^{2+}$ Ringer's solution containing (in mM) 155 NaCl, 4.5 KCl, 2 CaCl$_2$, 1 MgCl$_2$, 10 D-glucose, and 5 HEPES (pH 7.4). T cells were depolarized by perfusion with Ringer's solution containing (in mM) either 60 KCl, 95 NaCl, 2 CaCl$_2$, 1 MgCl$_2$, 5 HEPES and 10 D-glucose or 150 KCl, 2 CaCl$_2$, 1 MgCl$_2$, 5 HEPES and 10 D-glucose. Ca$^{2+}$ levels in T cells were analyzed by measuring either the peak of F340/380 ratios or the area under the curve (AUC) for a specific time period as indicated in each experiment.

For [Ca$^{2+}$]$_i$ measurements using a FlexStation 3 Multi-Mode Microplate Reader (Molecular Devices), T cells were loaded with Fura-2 as described above, resuspended in 40 µl Ringer's solution containing 2 mM CaCl$_2$ and plated into 96-well plates (Falcon, 353219) pre-coated with 0.01% poly-L-lysine. To depolarize T cells, an equal volume of Ringer's solution containing (in mM) 120 KCl, 35 NaCl, 2 CaCl$_2$, 1 MgCl$_2$, 5 HEPES and 10 D-glucose was added to achieve a final KCl concentration of 60 mM. For stronger depolarization, an equal volume of Ringer's solution containing (in mM) 300 KCl, 2 CaCl$_2$, 1 MgCl$_2$, 5 HEPES and 10 D-glucose was added to achieve a final KCl concentration of 150 mM. The resulting membrane potentials were calculated using the Nernst equation. Changes in Fura-2 ratios (F340/380) were analyzed using the SoftMax Pro 5.4.6 software (Molecular Devices). For some experiments, T cells were activated by TCR stimulation as follows. Mouse T cells were incubated with 1 µg/ml biotin-conjugated anti-CD3ε Ab (145-2C11; BD pharmingen, 553059) at the time of Fura-2 loading and stimulated by addition of 1 µg/ml streptavidin (Invitrogen, SNN1001) during Ca$^{2+}$ measurements. Human T cells were stimulated with 5 µg/ml anti-CD3 monoclonal antibody OKT3 during Ca$^{2+}$ measurements. At the end of some experiments, T cells were stimulated with 1 µM ionomycin. For some experiments, T cells were pre-incubated with 200 nM PMA or 1 mM 8-Bromoadenosine 3′,5′-cyclic monophosphate (8-Br-cAMP, Sigma-Aldrich, B7880) for 10 min before Ca$^{2+}$ measurements.

**Patch-clamp electrophysiology**. Patch-clamp experiments were conducted in the standard whole-cell recording configuration at room temperature using an Axopatch 200 amplifier (Axon Instruments) interfaced to an ITC-16 input/output board (Instrutech) and a Macintosh G3 computer. Recording electrodes were pulled from 100 µl pipettes coated with Sylgard and fire-polished to a final resistance of 2–5 MΩ. Stimulation and data acquisition and analysis were performed using in-house routines (R. Lewis, Stanford University) developed on the Igor Pro platform (Wavemetrics). Currents were filtered at 1 or 2 kHz with a 4-pole Bessel filter and sampled at 5 kHz. Data are corrected for the liquid junction potential of the pipette solution relative to Ringer's in the bath (−10 mV). Unless noted otherwise, all data were corrected for leak currents collected in Ca$^{2+}$-free Ringer's solution (for CRAC currents) or using the $P/8$ method[37] for VGCC currents, and averaged results are presented as the mean ± SEM. Curve fitting was done by least-squares methods using built-in functions in Igor Pro 4.0. For simultaneous measurements of [Ca$^{2+}$]$_i$, human T cells were loaded with 1 µM Indo-1-AM in culture medium at 37 °C for 15 min, washed and plated onto coverslips on the stage of a Nikon Diaphot TMD microscope. Cells were illuminated for 50–100 ms at 360 nm (360/25 filter; Chroma Technology) through a 40X Nikon Fluor objective (NA 1.3), and the fluorescence emissions at 405 nm and 485 nm (405/25 and 485/25 filters; Chroma Technology) were collected simultaneously with two photomultipliers (Hamamatsu). [Ca$^{2+}$]$_i$ was estimated from the relation [Ca$^{2+}$]$_i$ = K* (R-R$_{min}$)/(R$_{max}$-R). K*, R$_{min}$, and R$_{max}$ were measured in control human T cells in situ as previously described[100]. Ca$^{2+}$-free Ringer's solution was prepared by substituting 1 mM EGTA + 2 mM MgCl$_2$ for CaCl$_2$. For measurements of $I_{CRAC}$ and VGCC Ca$^{2+}$ currents using the perforated-patch configuration, 20 mM CaCl$_2$ was used. The isotonic Ba$^{2+}$ solution contained 110 BaCl$_2$, 10 HEPES, and 10 D-glucose. The Δ[Ca$^{2+}$]$_i$ in cells stimulated with a train of depolarizing pulses was determined from the difference in [Ca$^{2+}$]$_i$ immediately prior to the first +10 mV pulse and following termination of the pulse train. For the Δ[Ca$^{2+}$]$_i$ rises evoked by TG-activated CRAC currents in response to −100 mV hyperpolarizing pulses, Δ[Ca$^{2+}$]$_i$ was determined as the difference in [Ca$^{2+}$]$_i$ following re-addition of extracellular Ca$^{2+}$ (20 mM) for 30 s. Internal patch-clamp solutions varied with the identity of the current being studied and had the following compositions (in mM): $I_{CRAC}$: 150 Cs-methanesulfonate, 8 MgCl$_2$, 10 BAPTA, and 10 Cs-HEPES (pH 7.2). L-type Ca$^{2+}$ current: 115 Cs-aspartate, 1 CaCl$_2$, 5 MgCl$_2$, 10 HEPES, and 10 NaCl. All internal solutions had a pH of 7.2. Thapsigargin (TG, LC Biochemicals) was diluted from a 1 mM stock in DMSO to a final concentration of 1 µM. The extracellular solution contained charybdotoxin (Sigma) at a final concentration of 2 nM. Solutions were applied to the cells using a multi-barrel local perfusion pipette with a common delivery port. The time for 90% solution exchange was measured to be <1 s, based on the rate at which the K$^+$ current reversal potential changed when the external [K$^+$] was switched from 2 mM to 150 mM. For VGCC current recordings in mouse T cells and HEK293 cells, the internal solution was (in mM): 135 Cs aspartate, 8 MgCl$_2$, 8 Cs-BAPTA, and 10 HEPES, pH 7.2 with CsOH. The extracellular solution contained (in mM): 130 NaCl, 4.5 KCl, 20 CaCl$_2$, 10 tetra-ethylammonium chloride, 10 D-glucose, and 5 HEPES, pH 7.4 with NaOH. For Kv current recordings, the internal solution contained (in mM): 125 K MeSO$_4$, 6 MgCl$_2$, 10 EGTA, and 10 HEPES, pH 7.2 with KOH. The extracellular solution for these experiments contained (in mM): 145 NaCl, 4.5 KCl, 2 CaCl$_2$, 1 MgCl$_2$, 10 D-glucose, and 5 HEPES. pH was adjusted to 7.4 with NaOH.

**Immunoblotting**. Total cell lysates were prepared on ice using lysis buffer containing (in mM) 150 NaCl, 10 EDTA, 10 EGTA, 50 Tris, 1% Triton X-100, and 1% protease inhibitor cocktail (Sigma, P8340). For some protein extractions, lysis buffer was supplemented with 1% Nonidet P-40 (BioVision, 2111-100) and defined proteinase inhibitors: 10 µg/ml pepstatin A (Sigma-Aldrich 516481), 1 µg/ml leupeptin (Sigma-Aldrich L2884), 2 µg/ml aprotinin (Sigma-Aldrich A1153), 200 nM phenylmethanesulfonyl fluorid (PMSF; Millipore Sigma 10837091001). Cell debris

was removed by centrifugation at $15,000 \times g$ for 5–30 min and protein extracts were treated with Laemmli sample buffer (5% β-mercaptoethanol, 0.01% bromophenol blue, 10% glycerol, 2% SDS, 63 mM Tris-HCl, and 1% protease inhibitor cocktail, pH 6.8, all from Sigma), heated at 65 °C for 15 min (for Cav1.2, Cav1.3 and Cav1.4) or not (for Cavβ1), and spun down at $15,000 \times g$ for 30 s. Protein extracts were separated by SDS-PAGE using Novex™ WedgeWell™ 4–20% tris-glycine mini protein gel (Invitrogen, XP04205BOX) and transferred to polyvinylidene difluoride membranes. After blocking, membranes were incubated with custom-made antibodies that recognize endogenous Cav1.2, Cav 1.3 and Cav1.4 proteins. For detection of Cav1.2, a rabbit polyclonal antibody (FP1, 1:2000, polyclonal, provided by J.W.H.) that targets a residue between amino acids (aa) 783-845 in the intracellular loop between domains II and III at the N-terminus of rat neuronal Cavα1.2 (SPEKKQEVMEKPAVEESKEEKIELKSITADGESPPTTKINMDDLQPSE-NEDKSPHSNPDTAGE) was used[101]. For detection of Cav1.3, a rabbit polyclonal antibody (1:500 to 1:1000 dilution, provided by A.L.) that recognizes an N-terminal epitope of Cav1.3 protein (MQHQRQQQEDHANEANYARGTRKC) was used[102]. For detection of Cav1.4, a rabbit polyclonal antibody (1:500 to 1:1000 dilution, provided by A.L.) that recognizes an N-terminal epitope of the mouse Cav1.4 protein (MSESEVGKDTTPEPSPANGTC) was used[103]. Cavβ1 protein was detected using a mouse monoclonal antibody (1:600 dilution, Abcam, ab85020) that recognizes a conserved peptide in rat and mouse Cavβ1 (PMEVFDPSPQGKYSKR). Mouse anti-β-Actin monoclonal antibody (1:5,000 dilution, Proteintech, 66009-1-Ig), mouse anti-GAPDH (1:4,000 dilution, Cell Signaling, 14C10) and rabbit anti-Vinculin (1:2,000 dilution, CST, 4650) were used as loading controls. Membranes were incubated overnight with primary antibodies at 4 °C and immunoreactive bands were detected after incubation with HRP-conjugated secondary antibody (1:5000 to 1:10,000, Sigma, A9044) for 2 h at RT, and visualized by enhanced chemiluminescence using Clarity Max Western ECL Substrate (Bio-Rad, 1705062) or Classico Western HRP substrate (Millipore, WBLUC0100) using an Amersham 680-GE system. Alternatively, detection of β-Actin was performed after incubation with IRDye 680RD donkey anti-mouse IgG secondary antibody (1:10,000, LI-COR, 925-68072) for 1 h at RT, with an Odyssey Fc Western Blot Detection System (Licor Bioscience). Band densities were quantified and analyzed using ImageJ 1.52a.

**Real-time quantitative PCR.** Total RNA was isolated from T cells using TRIzol™ Reagent (Invitrogen, 15596026) and quantified using a Nanodrop 8000 spectrophotometer (Thermo Scientific). cDNA was synthesized using the iScript™ cDNA synthesis kit (Bio-Rad, 1708890). Quantitative real-time PCR was performed using the Maxima SYBR Green qPCR Master Mix (Thermo Scientific, K0223). Transcripts levels were normalized to the expression of housekeeping genes using the $2^{-\Delta CT}$ method. Quantitative Real-Time PCR data was acquired using QuantStudio Design & Software Analysis in a QuantStudio 3 PCR machine (Applied Biosystem, Thermo Fisher Scientific). The complete list of primers used can be found in Supplementary Table 4.

**Statistical analysis.** Data are expressed as mean ± SEM. Variables normally distributed according to according to Anderson–Darling, Kolmogorov–Smirnov or Shapiro–Wilk tests were analyzed by Student's $t$ test for comparisons between two groups. When the variables were not normally distributed, statistical significance was determined by using the non-parametric Mann–Whitney U test or the Kruskal–Wallis test. Statistical analyses were performed using GraphPad Prism 9 software. A value of two-tailed $P < 0.05$ was considered statistically significant.

**Reporting summary.** Further information on research design is available in the Nature Research Reporting Summary linked to this article.

## Data availability

The RNA-Seq data generated for this study have been deposited in the GEO database (GSE179625) at. Additional, published gene expression datasets analyzed in this study were downloaded from GEO and are listed in Supplementary Table 2 with information including accession numbers, cell types, species and PMID numbers[47,94,95,104–115]. Source data are provided with this paper.

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

## Acknowledgements

This work was funded by National Institutes of Health (NIH) grants AI097302, AI125997 and TR002873 to S.F. and an Irma T. Hirsch career development grant to S.F., NIH grant GM45374 to R.S.L., NIH grant NS057499 to M.P., NIH grants EY026817 and TR002916 to A.L., NIH grant AG055357 to J.W.H. Additional funding came from the Alfonso Martin Escudero Foundation (postdoctoral fellowship to A.R.C.), NYU (Bernard Levine postdoctoral fellowship to A.R.C.) and the Deutsche Akademie der Naturforscher Leopoldina (postdoctoral fellowship to S.E.) and a pre-doctoral fellowship by the T32 training program in Immunology and Inflammation (AI100853) to A.Y.T. We acknowledge the help of the Memorial Sloan Kettering Cancer Center (MSKCC) Gene Editing & Screening Core Facility, which is partially supported by a Cancer Center Support Grant NIH (P30-CA008748), for help with the design of the shRNA library. We thank S. Kahlfuss for preparation of human CD4+ T cells for RNA sequencing, J. Hagen (U. Iowa) for assistance with Cav1.4 Western blots, P. Schwarzberg (NIH) for providing pMRI-Amt and pMRI-GFP plasmids for the expression of guide RNAs, R.W. Tsien (New York University) for providing overexpression plasmids for rCav1.2, α2δ1 and β2α, and M. Pipkin (Scripps Research Institute) for providing pLMPd-Amt and pLMPd-GFP plasmids for expression of shRNAs and the ecotropic packaging vector pCL-Eco.

## Author contributions

S.E., A.R.C., R.S.L., M.P., and S.F. designed the research. S.E., A.R.C., W.L., B.L., M.Y., M.P., W.L., B.H., and B.L. performed experiments. A.R.C., A.Y.T., I.S., and P.P.R. performed bioinformatic analyses. R.G., A.L., and J.W.H. provided reagents. S.E., A.R.C., I.S., A.Y.T., W.L., P.P.R., A.L., J.W.H., M.Y., R.S.L., M.P., and S.F. analyzed the data. S.E., A.R.C., M.P., and S.F. wrote the paper with additional input by J.W.H.

## Competing interests

S.F. is a cofounder of CalciMedica. All other coauthors declare no competing interests.
