## [Peer Review File · Nature Communications]

REVIEWER COMMENTS

Reviewer #1 (Remarks to the Author):

This manuscript addresses an important and controversial subject. The manuscript takes on a long-standing controversy in the field of ion channels in T cells: are voltage-gated Ca²⁺ channels (VGCC) expressed in T cells? Do they function as channels to generate calcium influx? And, do they have functional roles aside from ion channel activity? This somewhat thankless task (refuting literature with experiments that did not show functional VGCC) is accomplished through a combination of novel molecular and expertly performed single-cell physiological approaches. Regardless of what the beta subunits are doing, the finding that human and mouse T cells do not have functional voltage-gated Ca²⁺ channels is very important, and the approach of failing to detect is rigorously applied. Since the conclusion relies on a set of negative findings (i.e. no detection), the presentation would benefit from positive control experiments, as suggested below in points #1 and 2. The experiments are convincing and contribute importantly toward resolving the issue.

The results introduce a novel screening approach to identify ion channels and transporters expressed in human and mouse T cells. The list of expressed genes is a gold-mine of information. Among molecular components of bona fide VGCCs, several different alpha pore forming and beta auxiliary subunits were identified. Among these, Cavb1 was targeted by CRISPR/Cas9 gene editing or by shRNA to knock down expression in antigen-specific mouse T cells from SMARTA mice. Transfer of T cells followed by LCMV virus infection identified genes that had effects on T cell clonal expansion in vivo. In particular, deletion of the gene encoding the CaVb1 subunit significantly reduced cell expansion during infection by inducing apoptosis, without effects on the ability of cells to proliferate or to produce cytokines (IL-2, TNF- α , IFN- γ). Importantly, deletion of the b1 subunit had no effect on TCR-induced Ca²⁺ influx or store-operated Ca²⁺ entry.

The manuscript examines whether mouse or human CD4⁺ T cells express VGCC channel activity using a variety of single-cell measurements. High external K⁺-induced depolarization was used to test for an expected rise in cytosolic Ca²⁺ (if VGCC channels were activated by depolarization). This maneuver had no effect. The experiments using combined perforated-patch recording and simultaneous Ca²⁺ monitoring and showed that depolarization did not produce VGCC current (using Ba²⁺ or Ca²⁺ outside) or rise in cytosolic Ca²⁺. These experiments are expertly and convincingly done.

The manuscript goes on to consider whether TCR stimulation or activation of PKA or PKC might be required to see VGCC activity, but again they don't detect it. Finally, the manuscript examines effects of STIM and Orai1 deletion in human T cells, to see if loss of function might reveal VGCC activity, but neither one does. This is a great approach because it uses human T cells from patients with loss of function mutations.

Finally, with respect to the expression of various alpha pore subunits, the manuscript shows that several subunits of VGCC are expressed at the RNA level, but not at the protein level (by antibodies "with confirmed specificity"). The RNA-seq exon analysis showed that transcripts were truncated, providing an explanation for why neither channel activity nor proteins were found. One wonders how or why, but ok.

Questions and comments 1 and 2 below include a recommendation for positive controls done under the exact same conditions but using a cell that does express voltage-gated Ca²⁺ channels. Comment 3 below asks if experiments on Jurkat T cells could be done.

1) Section starting on p. 10 with high K induced depolarization would benefit from control experiments performed on cells that do express functional VGCC currents. The high-K experiments should indeed induce a rise in cytosolic Ca²⁺ if voltage-gated Ca²⁺ channels are active, but under

the conditions used here do the protocols work in cells that express voltage-gated Ca²⁺ channels. One might expect a transient increase in Ca²⁺ if Ca_v channels that inactivate are involved. The positive control under the same conditions using a cell type that does express VGCC would help. The work of Carol Deutsch should be cited in discussing the results (Deutsch C, Price M. Role of extracellular Na and K in lymphocyte activation. *J Cell Physiol* 1982;113:73–79).

2) The experiments starting on p. 11 with perforated-patch recordings used Cs⁺ inside and charybdotoxin outside at very high concentrations, presumably to block both outward and inward currents through K⁺ channels. Could inward currents through Kv1.3 channels be evoked by depolarization under these conditions if charybdotoxin were omitted? It would be nice to provide a rationale for using such a high concentration of charybdotoxin (2 micromolar). A set of positive control experiments using a cell type that is known to have activatable VGCC channels would again be welcome here; the experiments should be done exactly under the same conditions. Also, please clarify how many cells, statistics, expectations of how many channels? How many VGCC would correspond to the detection limit of a 4 nM rise in Ca²⁺ with 790 fC of Ca influx? Where did the charge estimate come from, does the estimate include buffering?

3) Although the authors wisely do not chase every report in the literature, there are several reports of voltage-gated inward currents in Jurkat T cells that sort of look like voltage-gated calcium current (for example Dupuis et al., *J Physiol* 1989 PMID: 2557424). Perhaps this was part of the motivation for using charybdotoxin to block inward current through Kv1.3? Then, there are the high profile back to back Science papers from Ricardo Dolmetsch and Don Gill about STIM1 suppressing VGCC (PMID: 20929812 and 20929813). The Dolmetsch paper included patch-clamp experiments that provided motivation for the experiments on STIM1-deficient patients (Fig. 6G), and was referred to in the Discussion on p. 19: "The deletion of STIM1, which was reported to inhibit Cav1.2 in Jurkat T cells (40), did not evoke Ca²⁺ influx upon depolarization of mouse or human T cells." The Gill paper (no patch clamp) was not cited. Jurkat cells are not the same as human or mouse T cells, yet a lot of research on T cell activation has relied upon Jurkats. So, I wonder whether the authors have tried to repeat the observation about STIM1 knock-down in Jurkat cells, and if they can clear up some of the literature by further Discussion. I understand that the focus of this study is human and mouse T cells. Nor do they exclude the possibility that some subsets might express functional VGCCs.

Minor comments:

Pharmacology. Please be careful not to perpetuate the mistaken notion that CCBs are specific and exert their effects on T cells by blocking VGCC. p. 4: "Ca²⁺ channel blocker targeting VGCCs such as nimodipine, verapamil and diltiazem ,,,"; p12 "nimodipine, a potent blocker of L-type VGCCs."; p. 18 "Evidence supporting a role of VGCCs in T cells comes from the use of dihydropyridine Ca²⁺ channel blockers, RNAi mediated knockdown of VGCC expression and knockout mice (15, 17, 19, 20, 23, 24, 53-55)." These agents also block Kv1.3, as do the polyvalent cation blockers used by some studies as support for VGCC (DeCoursey, *J Neuroimm*, 1985 PMID: 2414315).

P. 5: "A role of VGCCs in T cells, however, is not universally accepted, and biophysical evidence of VGCC currents in T cells is limited (17, 19)". The citation is only to the limited evidence part of the sentence, not to the first part of the sentence. Recommend citing Cahalan and Chandy 2009 review for the first part of the sentence; this review discusses why it is not universally accepted.

p. 5 last sentence: "We conclude that although several α_1 and β subunits of VGCCs are expressed in T cells, they do not function as canonical VGCCs". The way this is worded it seems to contradict one of the main findings, that the incomplete transcripts do not lead to functional protein. The word "expressed" is ambiguous.

P 12 middle: "We next directly measured VGCC currents in human T cells after TCR stimulation with OKT3 (Figure 5G,H)". The sentence implies that you measured VGCC; but no, you did not!

p. 15 top: "monoclonal antibodies with confirmed specificity"; confirmed how?

Discussion p. 18: "prior stimulation of T cells by TCR crosslinking..." Raises the question of whether

activated T cell blasts or any number of T cell subsets might express VGCC. Was this evaluated in activated T cells or in subsets?

p. 19 bottom: "Micromolar concentrations of VGCC blockers, however, also inhibit several K⁺ channels," recommend adding "notably Kv1.3 channels in T cells"

p. 22 near top: "Strong genetic of electrophysiological evidence of functional ..." or?

Reviewer #2 (Remarks to the Author):

The manuscript reports an shRNA screen to identify channel and transporter genes that support T cell expansion in vivo during an LCMV infection. *Cacnb1* (encoding Cav β 1) is identified in the screen and validated as a positive regulator of T cell expansion. However, *Cacnb1* is found to be dispensable for TCR-elicited calcium influx and for upregulation of some key cytokines in vitro. Further, no voltage-dependent calcium current is detected in T cells, even after TCR stimulation, casting further doubt on whether Cav β 1 in T cells has its conventional role as a calcium channel subunit.

Here the manuscript pivots to the question of whether Cav α 1 subunits are expressed at all in T cells. The foregoing conclusions would be unchanged whether the answer is yes or no, but, having raised the issue, the manuscript falls short by failing to address some of the best documented claims in the literature that Cav α 1 subunits are present and functional in T cells (see comments (6)-(8), below).

(1) The manuscript gains its cachet from the in vivo shRNA screen, and Figure 1B makes it evident that several genes other than *Cacnb1* scored as hits in the screen. These genes should be named, even if they are not investigated further in this report.

(2) A conceivable criticism is that the study uncovers *Cacnb1* as a positive regulator of T cell expansion and survival in LCMV infection, then does not pinpoint an actual mechanism explaining its action. However, a full presentation of the shRNA screen and the genes that scored as hits would argue in favor of publication, even in the absence of a defined mechanism for *Cacnb1*.

(3) The most persuasive results reported are that Cav β 1 is not needed for a robust calcium response to TCR stimulation nor for calcium-dependent induction of the cytokines TNF, IFN γ , and IL-2. Of course this finding does not preclude Cav β 1 involvement in the induction of other cytokines, or in other calcium-dependent processes.

(4) It is also clear that a voltage-dependent calcium current is not detected under standard recording conditions in the T cells examined.

(5) Some readers will immediately grasp the argument regarding the sensitivity of Indo-1 as a measure of calcium influx: that 790 fC calcium influx in a 100-ms time window can be detected as a 4 nM increase in cytoplasmic calcium. Others will not. The text should explain clearly that the conclusion derives from the integrated current at each 100-ms voltage step to -100 mV and the slope of the calculated calcium concentration plot in Figure 4F.

(6) *Cacna1f* (Cav1.4). Omilusik et al., ref. 17, observed Ba²⁺ currents under voltage-step and voltage-ramp conditions suited to measure L-type calcium currents, in naive wild-type CD4⁺ and CD8⁺ T cells, but not in *Cacna1f*^{-/-} T cells. Because the current study did not examine mouse T cells by electrophysiology, it is unclear whether the discrepancy reflects a mouse-human difference, a naive T cell-experienced T cell difference, or an outcome of carrying out the experiments at different times in different laboratories. At a minimum, the authors should test electrophysiologically for Ba²⁺ current in mouse naive T cells, and examine directly by RNA-seq whether *Cacna1f* mRNA (or mRNA encoding another α 1 subunit) is expressed in mouse naive T cells. Retrieving data from the GEO database will not meet the need to test current and mRNA expression in identical cells.

(7) *Cacna1s* (Cav1.1). Matza et al., ref. 44, cloned full-length (alternatively spliced) CACNA1S / *Cacna1s* cDNAs from human Jurkat and mouse DO11.10 T cells. The evidence comes with the

caveat that these were T cell lines, but it calls for close investigation of the possibility that the full-length mRNA is expressed in primary T cells. The manuscript should provide evidence on whether full-length CACNA1S / *Cacna1s* cDNAs are present in the human and mouse T cells used for the experiments.

(8) *Cacna1g* (*Cav3.1*). Wang et al., ref. 19, reported finding *Cacna1g* mRNA and *Cav3.1α* protein in wild-type but not in *Cacna1g*^{-/-} CD4⁺ T cells, with a corresponding low voltage-activated current. The knockout cells exhibited no deficit in calcium signaling triggered by anti-CD3. Again, the manuscript needs to provide mouse RNA-seq data for the T cells used in this study.

(9) This has been a controversial area, and the present study is unlikely to be the last word on *Cav* channels in T cells. Nonetheless, it meets a high technical standard, and it will be a solid contribution to the literature on T cell calcium signaling if the above issues are addressed.

Reviewer #3 (Remarks to the Author):

The authors present an extremely interesting account of the possible involvement of the CaV-β1 subunit in the function of T cells. They show that CaV-β1 functions as a novel regulator of clonal expansion of T cells. The paper is particularly elegant using an shRNA-screening approach to identify ion channels that control T cell-mediated immunity. These studies revealed that CaV-β1 is required for the clonal expansion of CD4⁺ T cells following in vivo LCMV infection. The authors show that CaV-β1 is required to prevent T cell apoptosis following TCR stimulation in vitro, although elimination of CaV-β1 did not alter the proliferation of T cells.

The paper is really focused on understanding why the CaV-β1 should be having such effects on T cells and exploring the obvious hypothesis that this would be through altered function and/or expression/location of voltage-gated Ca²⁺ channels. Indeed, the authors present a very comprehensive description of the many studies that have reported important actions and possible roles of VGCCs in T cells. Moreover, there have been quite a number of studies reporting the action of the different CaV-β subunits within T cells. Thus, the other three β subunits, CaV-β2, CaV-β3 and CaV-β4 have all been implicated in regulating T cells in previous papers, and in each case these actions have been ascribed to their modification of VGCCs and alteration of Ca²⁺ signals. So it was surprising when the authors of the current paper began to reveal data the indicated the actions of CaV-β1 are likely independent of any effects on VGCCs.

In addition to revealing the interesting requirement of the CaV-β1 subunit on T cell function, the strength of the paper rests to the thoroughness and comprehensives of the many studies revealing that the effects of CaV-β1 are not through VGCCs. These experiments are compelling. Thus, they show that depolarization of T cells by patch-clamp analysis did not activate VGCC currents even under optimal recording conditions, that there were no effects of TCR activation on VGCC currents, that there were no VGCC currents in T cells lacking *Orai1* channels, that depolarization of T cells failed to induce Ca²⁺ influx in mouse or human T cells, and that deletion of *STIM1* which can inhibit VGCCs, did not give rise to any VGCC-mediated Ca²⁺ entry. Based on these rather comprehensive analyses they conclude that there is no evidence of functional VGCCs in T cells. All these studies are extremely carefully conducted. The only slight criticism would be that it might have been better to try to eliminate both *Orai1* and *STIM1* proteins at the same time in T cells. Elimination of only *Orai1* could have possibly increased the inhibitory effects of endogenous *STIM1* on CaV channels, and elimination of *STIM1* alone might have effects masked by the presence of *Orai1* and possibly *STIM2*. However, this is a minor point and the overall conclusion of all the studies is compelling.

The paper derives perhaps most of its significance from showing that there are no functional VGCCs in T cells. This conclusion is as important, perhaps more so, that revealing the action of CaV-β1 on T cell function. Thus, in the earlier papers, functional VGCCs have been claimed to be

present in T cells and exert effects on Ca²⁺ signals and T cell function. Moreover, the earlier studies on other CaV- β subunits and their role in T cell development/survival/function have been ascribed to their role in controlling function and/or expression of VGCCs. Whereas, the negative results on VGCC involvement in the current paper could be argued as less compelling than the "positive" role of VGCCs in T cells in the earlier papers, the current paper carries with it one additional and persuasive punch. Thus, the authors show that although T cells do contain VGCC transcripts and even perhaps protein, their expression profile related to splice variations present, indicates that they are not full-length or functional proteins. Certainly the evidence for $\alpha 1$ pore units lacking an N-terminal region are exclusively expressed in T cells is important. Also, the authors provide good evidence that splice changes in other VGCCs render their expression of functional channels quite unlikely. So this provides a compelling, albeit provocative, message for the paper since it indicates that the rather small actions of VGCCs on Ca²⁺ changes in T cells may be spurious.

Whereas all the above provides quite compelling justification for the conclusion of the paper, one could argue that, given the apparent significance of the CaV- $\beta 1$ subunit in clonal T cell expansion after viral infection and also in T cell apoptosis, it would have enhanced the impact of the paper if it had also included some mechanistic insights on the possible actions of CaV- $\beta 1$. This is particularly true, since it is now being suggested that the actions of the other CaV- β subunits in T cells may also be independent of VGCCs. Perhaps there is some underlying role of all these subunits? Perhaps a direct comparison of the role of CaV- $\beta 1$ with one or more other CaV- β subunits would be enlightening. Otherwise, it could be argued that the current paper is just adding on to the already described roles of other CaV- β subunits.

Overall, this reviewer believes that the paper, even without mechanistic studies, does provide an important advance in rethinking the role of VGCCs in T cell function. And, as the authors state, the conclusion that the lack of functional VGCCs in T cells provides good evidence that VGCC blockers are unlikely to have off-target effects on immune cell function, is also one of some significance.

One more minor point. There is a lot of repetition from the results in the discussion. Although it would take some effort to combine these, it would significantly enhance the paper's readability.

Reviewer #1

This manuscript addresses an important and controversial subject. The manuscript takes on a long-standing controversy in the field of ion channels in T cells: are voltage-gated Ca²⁺ channels (VGCC) expressed in T cells? Do they function as channels to generate calcium influx? And, do they have functional roles aside from ion channel activity? This somewhat thankless task (refuting literature with experiments that did not show functional VGCC) is accomplished through a combination of novel molecular and expertly performed single-cell physiological approaches. Regardless of what the beta subunits are doing, the finding that human and mouse T cells do not have functional voltage-gated Ca²⁺ channels is very important, and the approach of failing to detect is rigorously applied. Since the conclusion relies on a set of negative findings (i.e. no detection), the presentation would benefit from positive control experiments, as suggested below in points #1 and 2. The experiments are convincing and contribute importantly toward resolving the issue.

The results introduce a novel screening approach to identify ion channels and transporters expressed in human and mouse T cells. The list of expressed genes is a gold-mine of information. Among molecular components of bona fide VGCCs, several different alpha pore forming and beta auxiliary subunits were identified. Among these, Cavb1 was targeted by CRISPR/Cas9 gene editing or by shRNA to knock down expression in antigen-specific mouse T cells from SMARTA mice. Transfer of T cells followed by LCMV virus infection identified genes that had effects on T cell clonal expansion in vivo. In particular, deletion of the gene encoding the CaVb1 subunit significantly reduced cell expansion during infection by inducing apoptosis, without effects on the ability of cells to proliferate or to produce cytokines (IL-2, TNF-a, IFN-g). Importantly, deletion of the b1 subunit had no effect on TCR-induced Ca²⁺ influx or store-operated Ca²⁺ entry.

The manuscript examines whether mouse or human CD4⁺ T cells express VGCC channel activity using a variety of single-cell measurements. High external K⁺-induced depolarization was used to test for an expected rise in cytosolic Ca²⁺ (if VGCC channels were activated by depolarization). This maneuver had no effect. The experiments using combined perforated-patch recording and simultaneous Ca²⁺ monitoring and showed that depolarization did not produce VGCC current (using Ba²⁺ or Ca²⁺ outside) or rise in cytosolic Ca²⁺. These experiments are expertly and convincingly done.

The manuscript goes on to consider whether TCR stimulation or activation of PKA or PKC might be required to see VGCC activity, but again they don't detect it. Finally, the manuscript examines effects of STIM and Orai1 deletion in human T cells, to see if loss of function might reveal VGCC activity, but neither one does. This is a great approach because it uses human T cells from patients with loss of function mutations.

Finally, with respect to the expression of various alpha pore subunits, the manuscript shows that several subunits of VGCC are expressed at the RNA level, but not at the protein level (by antibodies "with confirmed specificity"). The RNA-seq exon analysis showed that transcripts were truncated, providing an explanation for why neither channel activity nor proteins were found. One wonders how or why, but ok.

Questions and comments 1 and 2 below include a recommendation for positive controls done under the exact same conditions but using a cell that does express voltage-gated Ca²⁺ channels. Comment 3 below asks if experiments on Jurkat T cells could be done.

Response: We thank the reviewer for his/her thorough evaluation of our study and insightful summary and comments. We agree with the recommendation to use "positive controls done under the exact same conditions but using a cell that does express voltage-gated Ca²⁺ channels" and have conducted new experiments as discussed in detail below.

1) Section starting on p. 10 with high K induced depolarization would benefit from control experiments performed on cells that do express functional VGCC currents. The high-K experiments should indeed induce a rise in cytosolic Ca²⁺ if voltage-gated Ca²⁺ channels are active, but under the conditions used here do the protocols work in cells that express voltage-gated Ca²⁺ channels. One might expect a transient increase in Ca²⁺ if CaV channels that inactivate are involved. The positive control under the same conditions using a cell

type that does express VGCC would help. The work of Carol Deutsch should be cited in discussing the results (Deutsch C, Price M. Role of extracellular Na and K in lymphocyte activation. J Cell Physiol 1982;113:73–79).

Response: We thank the reviewer for this comment and agree that a positive control is required. We have used two approaches to address this suggestion: *First*, we have transfected HEK293 cells with the $\alpha 1$ pore subunit of the L-type Ca^{2+} channel Cav1.2 in combination with β , γ , and $\alpha 2\delta$ subunits to form a functional channel complex. These cells were subjected to the same protocol replacing Na^+ isotonicity with 150 mM K^+ as T cells. Exposure of Cav1.2 transfected HEK293 cells to 150 mM K^+ resulted in a transient increase in intracellular Ca^{2+} , which was not observed in untransfected HEK293 cells (**new Supplemental Figure 4A**). This increase could be suppressed by treating cells with 8 μM of the L-type Ca^{2+} channel blocker nimodipine (**new Supplemental Figure 4B**). We also directly measured voltage-gated Ca^{2+} currents in these cells. Stepwise depolarization of HEK293 cells transfected with Cav1.2 and its auxiliary subunits from -80 mV to +80 mV evoked robust Ca^{2+} currents that could be completely blocked with 10 μM nimodipine (**new Supplemental Figure 5A**). The *second* approach we took was to measure voltage-dependent Ca^{2+} influx and currents in excitable cells. To this end we used the rat pheochromocytoma cell line PC12. Exposure of PC12 cells to 150 mM K^+ resulted in a transient increase in intracellular Ca^{2+} (**new Supplemental Figure 4C**). Moreover, stepwise depolarization of PC12 cells from -80 mV to +60 mV from a holding potential of -70 mV elicited strong Ca^{2+} currents (**new Supplemental Figure 5B**). Collectively, these experiments demonstrate that we are able to record voltage-gated Ca^{2+} influx and currents in cells that express VGCCs either after overexpression or endogenously. We are therefore convinced that the absence of voltage-gated Ca^{2+} influx and currents in mouse and human T cells is not due to our recording conditions.

We cited the paper by Deutsch & Price (1982) in the Results section.

2) The experiments starting on p. 11 with perforated-patch recordings used Cs^+ inside and charybdotoxin outside at very high concentrations, presumably to block both outward and inward currents through K^+ channels. Could inward currents through Kv1.3 channels be evoked by depolarization under these conditions if charybdotoxin were omitted? It would be nice to provide a rationale for using such a high concentration of charybdotoxin (2 micromolar). A set of positive control experiments using a cell type that is known to have activatable VGCC channels would again be welcome here; the experiments should be done exactly under the same conditions. Also, please clarify how many cells, statistics, expectations of how many channels? How many VGCC would correspond to the detection limit of a 4 nM rise in Ca^{2+} with 790 fC of Ca influx? Where did the charge estimate come from, does the estimate include buffering?

Response: We thank the reviewer for his insightful comments. We indeed used charybdotoxin to block contaminating K^+ currents. The concentration mentioned in the text, however, was wrong. We used 2 nM (nanomolar), not micromolar, charybdotoxin. We apologize for the mistake, which has been corrected. We appreciate the question “Could inward currents through Kv1.3 channels be evoked by depolarization under these conditions if charybdotoxin were omitted?” This is another positive control to show that depolarization of T cells can evoke voltage-gated currents of channels that are expressed in T cells. Kv1.3 is an excellent example and we measured Kv1.3 in mouse T cells, either naive CD4^+ T cells freshly isolated from lymph nodes or T cells expanded for 3-5 days *in vitro* following activation by anti-CD3/CD28 stimulation. Depolarization of naive and activated T cells from -100 mV to +100 mV from a holding potential of -70 mV evoked robust K^+ currents under whole cell patch clamp recording conditions (**new Figure 5E,F**). Depolarization of the same naive or activated CD4^+ T cells from -80 mV to +60 mV, also in whole cell configuration, did not evoke any detectable voltage-gated Ca^{2+} currents (**Figure 5C,D**). Of note, the use of “more standard methods” (i.e. whole cell instead of perforated patch recordings) to measure VGCC currents was requested by reviewer 2.

We agree with the reviewer regarding the use of positive control cells. Please see our description of experiments using PC12 cells and HEK293 cells transfected with Cav1.2 above. For these experiments, we have used whole-cell patch clamp recordings to measure VGCC currents. We note that the simultaneous Ca^{2+} current and intracellular Ca^{2+} level recordings, while elegant, are not necessary to detect VGCC currents in excitable cells, and we think that the use of whole cell recordings is sufficient to show the presence of VGCC currents in excitable cells, and by comparison the lack of VGCC currents in murine T cells.

Regarding the question “How many VGCC would correspond to the detection limit of a 4 nM rise in Ca^{2+} with 790 fC of Ca influx? Where did the charge estimate come from, does the estimate include buffering?": The

detection of a rise in $[Ca^{2+}]_i$ in response to I_{CRAC} activation came from actual measurement, not an estimate. For the data shown in Figure 4F, the charge is the integral of the CRAC Ca^{2+} current that entered the cell during the step-ramp voltage protocol. This was then directly compared to the rise in $[Ca^{2+}]_i$ as detected by the Indo-1 signal. There are no estimates here, just a direct measurement. In the revised manuscript text we write on page 11 “In these latter experiments, a 6 nM rise in $[Ca^{2+}]_i$ immediately following readdition of extracellular Ca^{2+} could be detected with the 170 fC of Ca^{2+} influx that flowed through CRAC channels (assessed by integrating the Ca^{2+} current charge over the duration of the step-ramp pulse) (**Fig. 4G**).

3) Although the authors wisely do not chase every report in the literature, there are several reports of voltage-gated inward currents in Jurkat T cells that sort of look like voltage-gated calcium current (for example Dupuis et al., J Physiol 1989 PMID: 2557424). Perhaps this was part of the motivation for using charybdotoxin to block inward current through Kv1.3? Then, there are the high profile back to back Science papers from Ricardo Dolmetsch and Don Gill about STIM1 suppressing VGCC (PMID: 20929812 and 20929813). The Dolmetsch paper included patch-clamp experiments that provided motivation for the experiments on STIM1-deficient patients (Fig. 6G), and was referred to in the Discussion on p. 19: “The deletion of STIM1, which was reported to inhibit Cav1.2 in Jurkat T cells (40), did not evoke Ca^{2+} influx upon depolarization of mouse or human T cells.” The Gill paper (no patch clamp) was not cited. Jurkat cells are not the same as human or mouse T cells, yet a lot of research on T cell activation has relied upon Jurkats. So, I wonder whether the authors have tried to repeat the observation about STIM1 knock-down in Jurkat cells, and if they can clear up some of the literature by further Discussion. I understand that the focus of this study is human and mouse T cells. Nor do they exclude the possibility that some subsets might express functional VGCCs.

Response: We thank the reviewer for pointing out papers that have reported VGCC currents in T cells, which we now cite in the revised manuscript. Please note that we had cited both papers reporting an inhibitory effect of STIM1 on Cav1.2 as Ref. 39 (Wang/Gill 2010) and Ref. 40 (Park/Dolmetsch 2010). In our discussion we focused on the paper by the Dolmetsch lab, because it used Jurkat T cells to study the effects of STIM1 on VGCC function (unlike the paper from Don Gill’s lab, which used HEK293 cells). Importantly, Park et al only observed an effect of STIM1 deletion on depolarization-induced Ca^{2+} influx (with high K^+) when Cav1.2 was overexpressed in Jurkat cells. Jurkat cells that did not overexpress Cav1.2 did not respond with Ca^{2+} influx to depolarization with high K^+ (Figure 2C of Park et al, PMID 20929812). The results of these experiments therefore suggest that Jurkat T cells do not have endogenous VGCCs. (Note that we do not refute the findings of either the Dolmetsch or Gill lab paper that STIM1 inhibits Cav1.2, but it is important to emphasize that neither paper provides evidence for endogenous, functional VGCCs in Jurkat T cells). We clarified this point by revising the Discussion section. Regarding the use of Jurkat cells and the comment about “several reports of voltage-gated inward currents in Jurkat T cells that sort of look like voltage-gated calcium current”, we decided to draw the line somewhere and “not chase every report in the literature”. Jurkat cells are a human leukemic T cell line, which has indeed been used to great effect for elucidating signal transduction pathways in T cells. But even if these cells showed some evidence of voltage-gated Ca^{2+} influx or currents, what would be the significance of such a finding given that our data demonstrate the absence of functional VGCCs in mouse nor human primary T cells have functional VGCCs? (As an aside, we karyotyped several Jurkat cell strains we had in the lab many years ago and two of three were tetraploid instead of diploid. Cell lines such as Jurkats that are propagated for long periods of time in the lab are prone to acquiring mutations not present in primary T cells).

We do agree, however, with the last comment by reviewer 1 “Nor do they exclude the possibility that some subsets might express functional VGCCs”. Indeed, several labs have argued that functional VGCCs are present in Th17 cells (for instance Cav3.1 channels, Wang et al. 2016, PMID 27037192, or more recently Cav1.4 channels, Mars et al. 2021, PMID 34653514) or Th2 cells (for instance Robert et al. 2014, PMID 24365142 or Cabral et al. 2010, PMID 20167851). This is a valid concern because it is possible that specific VGCC channels are upregulated during the differentiation of $CD4^+$ T cells into distinct T helper subsets including Th2 and Th17 cells. To address this possibility, we first analyzed mRNA expression of α , β , γ and $\alpha\delta$ subunits in naive $CD4^+$ T cells and Th1, Th2, Th17 or Treg cells (**new Supplemental Figure 10A,B**). Again, *Cacna1a* (Cav2.1) was the most highly expressed α_1 subunit in $CD4^+$ T cells, with the highest levels in Treg cells. We paid specific attention to the expression of Cav1.2 and Cav3.1, which have been reported in Th2 and Th17 cells, respectively (see above). Compared to other $CD4^+$ T cell subsets, relatively higher mRNA levels

were observed for *Cacna1c* (Cav1.2) in Th2 cells and *Cacna1g* (Cav3.1) in Treg cells (**Supplemental Figure 10B**). We next tested whether the differentiation of naïve CD4⁺ T cells into these T cell subsets is associated with the occurrence of voltage-activated Ca²⁺ influx. The polarization of murine CD4⁺ T cells into Th2, Th17 and induced Treg (iTreg) cells was associated with the expected upregulation of lineage-specific transcription factors including GATA3 (Th2), Foxp3 (iTreg) and ROR γ t (Th17) (**Supplemental Figure 10C, D**). Depolarization of Th2, Th17 and iTreg cells with 60 mM or 150 mM K⁺ in the extracellular buffer, however, failed to evoke detectable Ca²⁺ influx (**Supplemental Figure 10E,F**). By contrast, ionomycin induced robust SOCE in all T cell subsets, which was suppressed by high extracellular K⁺, as expected. We conclude that VGCC function is undetectable in Th2, Th17 and iTreg cells.

Minor comments:

Pharmacology. Please be careful not to perpetuate the mistaken notion that CCBs are specific and exert their effects on T cells by blocking VGCC. p. 4: “Ca²⁺ channel blocker targeting VGCCs such as nimodipine, verapamil and diltiazem ,,,”; p12 “nimodipine, a potent blocker of L-type VGCCs.”; p. 18 “Evidence supporting a role of VGCCs in T cells comes from the use of dihydropyridine Ca²⁺ channel blockers, RNAi mediated knockdown of VGCC expression and knockout mice (15, 17, 19, 20, 23, 24, 53-55).” These agents also block Kv1.3, as do the polyvalent cation blockers used by some studies as support for VGCC (DeCoursey, J Neuroimm, 1985 PMID: 2414315).

Response: We agree with the reviewer and are well aware of the studies by Cahalan, Chandy and others who showed the effects of CCBs on K⁺ channels. We have elaborated in more detail on the unspecific effects of CCBs on other channels in more detail in the Discussion.

P. 5: “A role of VGCCs in T cells, however, is not universally accepted, and biophysical evidence of VGCC currents in T cells is limited (17, 19)”. The citation is only to the limited evidence part of the sentence, not to the first part of the sentence. Recommend citing Cahalan and Chandy 2009 review for the first part of the sentence; this review discusses why it is not universally accepted.

Response: We agree and have added a sentence to the Discussion and also cite the Cahalan / Chandy 2009 review.

p. 5 last sentence: “We conclude that although several α 1 and β subunits of VGCCs are expressed in T cells, they do not function as canonical VGCCs”. The way this is worded it seems to contradict one of the main findings, that the incomplete transcripts do not lead to functional protein. The word “expressed” is ambiguous.

Response: Thanks for catching this imprecise wording, which we corrected. The sentence now says “Collectively, these data demonstrate that although mRNAs for several VGCCs can be detected in mouse and human T cells, the transcripts are incomplete and result in N-terminally truncated proteins. [...]. Even if these proteins were stable and properly located in the plasma membrane, they would very likely not be functional Ca²⁺ channels, providing an explanation for the absence of VGCC currents and Ca²⁺ influx upon depolarization in T cells”.

P 12 middle: “We next directly measured VGCC currents in human T cells after TCR stimulation with OKT3 (Figure 5G,H)”. The sentence implies that you measured VGCC; but no, you did not!

Response: We agree of course and made the sentence more precise by writing “We *next attempted to* directly measure VGCC currents in human T cells ...”

p. 15 top: “monoclonal antibodies with confirmed specificity”; confirmed how?

Response: We have added sentences on pages 15 and 16, which explain how the specificity of antibodies against Cav1.2, Cav1.3 and Cav1.4 was determined.

Discussion p. 18: “prior stimulation of T cells by TCR crosslinking...” Raises the question of whether activated T cell blasts or any number of T cell subsets might express VGCC. Was this evaluated in activated T cells or in subsets?

Response: This description (“prior stimulation of T cells by TCR crosslinking”) refers to primary human T cells cultured in vitro and restimulated with anti-CD3 (OKT3) or mouse T cells that were stimulated with anti-CD3/CD28, expanded for several days in tissue culture restimulated by anti-CD3 crosslinking. No subsets of T cells (such as Th2 or Th17 cells) were used for current measurements. The expression levels of VGCC alpha subunits in human and mouse T cells without or with TCR crosslinking are shown in Figure 7A,B.

p. 19 bottom: “Micromolar concentrations of VGCC blockers, however, also inhibit several K⁺ channels,” recommend adding “notably Kv1.3 channels in T cells”

Response: We agree with this comment and have added the recommended subordinate clause to page 20 (top).

p. 22 near top: “Strong genetic of electrophysiological evidence of functional ...” or?

Response: Thank you for catching this mistake; we corrected “of” to “or”.

Reviewer #2

The manuscript reports an shRNA screen to identify channel and transporter genes that support T cell expansion in vivo during an LCMV infection. *Cacnb1* (encoding Cav β 1) is identified in the screen and validated as a positive regulator of T cell expansion. However, *Cacnb1* is found to be dispensable for TCR-elicited calcium influx and for upregulation of some key cytokines in vitro. Further, no voltage-dependent calcium current is detected in T cells, even after TCR stimulation, casting further doubt on whether Cav β 1 in T cells has its conventional role as a calcium channel subunit.

Here the manuscript pivots to the question of whether Cav α 1 subunits are expressed at all in T cells. The foregoing conclusions would be unchanged whether the answer is yes or no, but, having raised the issue, the manuscript falls short by failing to address some of the best documented claims in the literature that Cav α 1 subunits are present and functional in T cells (see comments (6)-(8), below).

(1) The manuscript gains its cachet from the in vivo shRNA screen, and Figure 1B makes it evident that several genes other than *Cacnb1* scored as hits in the screen. These genes should be named, even if they are not investigated further in this report.

Response: We understand the reviewer’s curiosity about other screen hits besides *Cacnb1* from the in vivo shRNA screen for ICTs that regulate antiviral immunity by T cells. Because these hits are not investigated further in this study and therefore not relevant for its conclusions, and because they are the topic of another study that we are currently preparing for publication, we decided to report the remaining screen hits in that manuscript and to not name them here. We ask the reviewer for his understanding of this decision.

(2) A conceivable criticism is that the study uncovers *Cacnb1* as a positive regulator of T cell expansion and survival in LCMV infection, then does not pinpoint an actual mechanism explaining its action. However, a full presentation of the shRNA screen and the genes that scored as hits would argue in favor of publication, even in the absence of a defined mechanism for *Cacnb1*.

Response: We agree with the reviewer that it would have been rewarding to “pinpoint an actual mechanism explaining its action” on T cell expansion during viral infection. This is exactly the reason why we went into such great detail to detect voltage-gated Ca²⁺ influx and currents in T cells, which is the canonical role of *Cacnb1* (Cav β 1), i.e. the regulation of the function of VGCCs in excitable cells. Specifically, we carefully investigated the main canonical activation pathways of voltage-gated calcium channels using different

approaches including Ca^{2+} imaging and patch-clamp electrophysiology. Neither approach, conducted on mouse and human T cells, under various conditions, was able to detect any evidence for VGCC function in T cells. In the absence of any detectable VGCC function in T cells, how *Cacnb1* regulates T cell apoptosis remains unclear. We comment on various potential mechanisms by which Cav beta subunits such as *Cacnb1* could affect T cell function. “A significant body of evidence demonstrates that they interact with many other proteins and have many VGCC independent functions^{7,25}. For instance, Cav β proteins interact with other ion channels including ryanodine receptors, membrane receptors, Ras-related monomeric small GTP-binding (RGK) proteins, dynamin, actin and the scaffolding protein AHNK1²⁵. One of the most intriguing functions of Cav β proteins is their role in controlling gene expression in the nucleus.” We discuss such roles (in the Discussion section), which have been reported for specific Cav β 4, Cav β 3 and Cav β 1a isoforms. It is therefore possible that Cav β 1 plays similar roles in T cells by controlling gene expression, which would be distinct from its regulatory role of VGCC function in excitable cells. We think, however, that finding out how *Cacnb1* (*Cav* β 1) regulates T cell expansion during viral infection independent of regulating VGCCs entails an open-ended, extensive investigation that is beyond the scope of this study, which is already rather comprehensive in scope.

Regarding the request for a “full presentation of the shRNA screen and the genes that scored as hits would argue in favor of publication, even in the absence of a defined mechanism for *Cacnb1*” does not help address the question how *Cacnb1* regulates T cell function because the screen hits are not functionally related and control T cell function independent of each other. As mentioned in response to the preceding critique, the other screen hits are the topic of another study that we are currently preparing for publication and will be reported there.

(3) The most persuasive results reported are that *Cav* β 1 is not needed for a robust calcium response to TCR stimulation nor for calcium-dependent induction of the cytokines TNF, IFN γ , and IL-2. Of course, this finding does not preclude *Cav* β 1 involvement in the induction of other cytokines, or in other calcium-dependent processes.

Response: We agree that in theory *Cacnb1* could be involved in the regulation of other cytokines or other calcium-dependent processes in T cells. We selected TNF, IFN γ , and IL-2 because of their expression is well known to be very dependent on Ca^{2+} signals downstream of TCR stimulation and even small reductions in Ca^{2+} influx (e.g. through CRAC channels) result in strong attenuation of TNF, IFN γ , and IL-2 expression. Therefore these cytokines are sensitive indicators of reduced Ca^{2+} signals in general. Since their expression was not impaired in *Cacnb1*-deficient T cells (compared to strong reduction in STIM1-deficient cells), it seems very unlikely that other cytokines are impaired or other Ca^{2+} dependent processes. The most important argument against such a role of *Cacnb1* in “the induction of other cytokines, or in other calcium-dependent processes”, however, is that Ca^{2+} influx was normal in *Cacnb1*-deficient T cells.

Although not specifically mentioned by Reviewer 2, we interpreted his/her comments to potentially suggest that *Cacnb1* might regulate VGCC function and Ca^{2+} signaling in other T cell subsets that produce other cytokines. As discussed in more detail in response to Reviewer 1, VGCCs have been implicated in Th17 cell function (Wang et al. 2016, PMID 27037192; Mars et al. 2021, PMID 34653514) and Th2 cells (Robert et al. 2014, PMID 24365142; Cabral et al. 2010, PMID 20167851). We therefore differentiated naïve CD4⁺ T cells into Th2, Th17 and Treg (regulatory T) cells and measured whether they have voltage-activated Ca^{2+} influx. Whereas the polarization of murine CD4⁺ T cells into Th2, Th17 and Treg cells was associated with the expected upregulation of lineage-specific transcription factors including GATA3 (Th2), Foxp3 (iTreg) and ROR γ t (Th17), their depolarization with 60 mM or 150 mM K⁺ in the extracellular buffer failed to evoke detectable Ca^{2+} influx (**Supplemental Figure 10C-F**). These experiments that VGCC are not functional in other T helper cell subsets such as Th2, Th17 and iTreg cells either. It is therefore very unlikely that *Cacnb1* is involved in Ca^{2+} dependent regulation of cytokine expression and we therefore did not further test this possibility.

(4) It is also clear that a voltage-dependent calcium current is not detected under standard recording conditions in the T cells examined.

Response: That is absolutely right and an important conclusion of our study.

(5) Some readers will immediately grasp the argument regarding the sensitivity of Indo-1 as a measure of

calcium influx: that 790 fC calcium influx in a 100-ms time window can be detected as a 4 nM increase in cytoplasmic calcium. Others will not. The text should explain clearly that the conclusion derives from the integrated current at each 100-ms voltage step to -100 mV and the slope of the calculated calcium concentration plot in Figure 4F.

Response: As correctly stated by the reviewer, the detection of a rise in $[Ca^{2+}]_i$ in response to I_{CRAC} activation came from direct quantification of the charge entering the cells (measured as the integral of the current over the 100 ms step-ramp pulse) and measurement of the corresponding rise in $[Ca^{2+}]_i$ as detected by the Indo-1 signal. We have now explained the rationale behind this argument more carefully in the manuscript text on page 11: "In these latter experiments, a 6 nM rise in $[Ca^{2+}]_i$ immediately following readdition of extracellular Ca^{2+} could be detected with the 170 fC of Ca^{2+} influx that flowed through CRAC channels (assessed by integrating the Ca^{2+} current charge over the 200 ms duration of the step-ramp pulse) (**Fig. 4G**)".

(6) *Cacna1f* (*Cav1.4*). Omilusik et al., ref. 17, observed Ba^{2+} currents under voltage-step and voltage-ramp conditions suited to measure L-type calcium currents, in naive wild-type $CD4^+$ and $CD8^+$ T cells, but not in *Cacna1f*^{-/-} T cells. Because the current study did not examine mouse T cells by electrophysiology, it is unclear whether the discrepancy reflects a mouse-human difference, a naive T cell-experienced T cell difference, or an outcome of carrying out the experiments at different times in different laboratories. At a minimum, the authors should test electrophysiologically for Ba^{2+} current in mouse naive T cells, and examine directly by RNA-seq whether *Cacna1f* mRNA (or mRNA encoding another $\alpha 1$ subunit) is expressed in mouse naive T cells. Retrieving data from the GEO database will not meet the need to test current and mRNA expression in identical cells.

Response: We thank the reviewer for these two comments, which we agree are important to address. (1) Regarding the possibility that the absence of VGCC currents in human T cells reported in our study compared to currents in mouse T cells reported by Omilusik et al.(ref. 17) could be a "mouse-human difference", we provide two arguments to support our conclusion that mouse T cells also lack functional VGCC channels. First, depolarization of mouse $CD4^+$ and $CD8^+$ T cells fails to elicit Ca^{2+} influx under any of the conditions we tested (e.g. Figure 5A,B). Second, in response to the reviewer's critique, we have now performed patch-clamp experiments in naive mouse T cells (**new Figure 5C**) and in activated mouse T cells (**new Figure 5D**). No inward Cav currents were detected in naive and activated T cells following stepwise depolarization from -80 to +60 mV from a holding potential of -70 mV. By contrast, another voltage-gated channels, the K^+ channel *Kv1.3*, which is well established to play a role in T cell function could be detected in mouse T cells. Depolarization of naive (**new Figure 5E**) and activated (**new Figure 5F**) T cells under voltage clamp conditions evoked robust K^+ currents.

(2) Regarding the second comment "examine directly by RNA-seq whether *Cacna1f* mRNA (or mRNA encoding another $\alpha 1$ subunit) is expressed in mouse naive T cells. Retrieving data from the GEO database will not meet the need to test current and mRNA expression in identical cells", we have analyzed our own RNA-Seq data we generated from $CD4^+$ T cells isolated from wildtype mice for the expression of VGCC $\alpha 1$ subunits and *Cav1.4* in particular. These data confirm the GEO-based expression data shown in Figure 7B and show that *Cacna1f* encoding *Cav1.4* is not expressed in murine T cells that were left unstimulated or stimulated for 24 or 48h. As in the GEO-based dataset, *Cacna1a* (encoding *Cav2.1*) was expressed in naive and activated T cells, whereas mRNA for *Cacna1i* (*Cav3.3*) was detectable only in naive T cells.

(7) *Cacna1s* (*Cav1.1*). Matza et al., ref. 44, cloned full-length (alternatively spliced) *CACNA1S* / *Cacna1s* cDNAs from human Jurkat and mouse DO11.10 T cells. The evidence comes with the caveat that these were

T cell lines, but it calls for close investigation of the possibility that the full-length mRNA is expressed in primary T cells. The manuscript should provide evidence on whether full-length CACNA1S / Cacna1s cDNAs are present in the human and mouse T cells used for the experiments.

Response: Thank you for the comment. To address this question, we checked the mRNA expression of Cacna1s (Cav1.1) in the same RNA Seq data generated in our lab from mouse CD4⁺ T cells described in the previous response. We are unable to detect any Cacna1s expression in either unstimulated T cells or T cells stimulated for 24 or 48h with anti-CD3/CD28 (see the **Figure above**). These results are consistent with the absence of Cacna1s expression in murine CD4⁺ and CD8⁺ T cells based on datasets retrieved from GEO (**Figure 7B**). Moreover, we did not detect CACNA1S mRNA expression in human T cells by RNA Seq, either unstimulated or stimulated T cells, from either healthy donors or a patient lacking STIM1 expression (**Figure 7A**). These are RNA Seq data generated in our own lab from primary human T cells cultured under the same conditions as those used for electrophysiology experiments in **Figure 4C-H**. We unambiguously conclude that neither mouse nor human primary T cells express CACNA1S (Cav1.1). These findings preclude us from testing if CACNA1S transcripts are full-length. As we had argued in response to Reviewer 1, Jurkat cells are a human leukemic T cell line, and even if these cells expressed CACNA1S in contrast to primary T cells from healthy donors or mice, what would be the significance of such a finding? We therefore decided to forego analyzing CACNA1S expression in Jurkat cells.

(8) Cacna1g (Cav3.1). Wang et al., ref. 19, reported finding Cacna1g mRNA and Cav3.1 α protein in wild-type but not in Cacna1g^{-/-} CD4⁺ T cells, with a corresponding low voltage-activated current. The knockout cells exhibited no deficit in calcium signaling triggered by anti-CD3. Again, the manuscript needs to provide mouse RNA-seq data for the T cells used in this study.

Response: Thank you for this comment. We checked the mRNA expression of Cacna1g (Cav3.1) in the same RNA Seq data generated in our lab from mouse CD4⁺ T cells described in the previous responses. We are unable to detect significant Cacna1g expression in either unstimulated T cells or T cells stimulated for 24 or 48h with anti-CD3/CD28 (see the **Figure above**). These results are consistent with the absence of Cacna1g expression in murine CD4⁺ and CD8⁺ T cells based on datasets retrieved from GEO (**Figure 7B**). It is noteworthy that Cav3.1 currents were detected by Wang et al. (2016) in mouse Th17 cells, and we therefore differentiated CD4⁺ T cells into Th17 cells and measured Ca²⁺ influx in response to depolarization with 60 mM or 150 mM K⁺. We failed to observe any Ca²⁺ increase upon depolarization in Th17 cells (**New Supplemental Figure 10E,F**). Positive controls, HEK293 cells transfected with Cav1.2 or PC12 cells that are electrically excitable, did show Ca²⁺ influx under the same recording conditions (**New Supplemental Figures 4 and 5**). For a more detailed description see our response to Reviewer 1. Together, we conclude that Th17 cells do not express significant levels of Cav3.1 that give rise to voltage-dependent Ca²⁺ influx.

(9) This has been a controversial area, and the present study is unlikely to be the last word on Cav channels in T cells. Nonetheless, it meets a high technical standard, and it will be a solid contribution to the literature on T cell calcium signaling if the above issues are addressed.

Response: Thank you for recognizing the technical tour-de-force that underlies this study, we appreciate it. We hope to have made an important contribution to the field by identifying a VGCC independent role of Cav β 1 (Cacnb1) in T cells and, as we think, raising the bar for those labs that want to claim that VGCC are functional in T cells. All but two papers (Omilusik 2011, Wang 2016) have failed to show VGCC currents in T cells, which in our opinion is a *conditio sine qua non* for proving that a channel is functional in a particular cell type.

Reviewer #3

The authors present an extremely interesting account of the possible involvement of the CaV- β 1 subunit in the function of T cells. They show that CaV- β 1 functions as a novel regulator of clonal expansion of T cells. The paper is particularly elegant using an shRNA-screening approach to identify ion channels that control T cell-mediated immunity. These studies revealed that CaV- β 1 is required for the clonal expansion of CD4⁺ T cells

following in vivo LCMV infection. The authors show that CaV-β1 is required to prevent T cell apoptosis following TCR stimulation in vitro, although elimination of CaV-β1 did not alter the proliferation of T cells.

The paper is really focused on understanding why the CaV-β1 should be having such effects on T cells and exploring the obvious hypothesis that this would be through altered function and/or expression/location of voltage-gated Ca²⁺ channels. Indeed, the authors present a very comprehensive description of the many studies that have reported important actions and possible roles of VGCCs in T cells. Moreover, there have been quite a number of studies reporting the action of the different CaV-β subunits within T cells. Thus, the other three β subunits, CaV-β2, CaV-β3 and CaV-β4 have all been implicated in regulating T cells in previous papers, and in each case these actions have been ascribed to their modification of VGCCs and alteration of Ca²⁺ signals. So it was surprising when the authors of the current paper began to reveal data that indicated the actions of CaV-β1 are likely independent of any effects on VGCCs.

Response: Thank you for the positive evaluation of our work, we appreciate it. The finding that Cavβ1 regulates T cell function independent of modulating Ca²⁺ influx in T cells came as a surprise to us, too, because of the previous studies showing a role of Cavβ2, β3 and β4 in T cell function and Ca²⁺ influx. Our findings then prompted us investigate the function of VGCCs in T cells more closely and after an exhaustive analysis of mouse and human T cells were able that VGCC function is not detectable in T cells. If VGCCs are not functional in T cells as Ca²⁺ channels, then it seems logical to us that Cavβ proteins must have other roles in T cells.

In addition to revealing the interesting requirement of the CaV-β1 subunit on T cell function, the strength of the paper rests to the thoroughness and comprehensives of the many studies revealing that the effects of CaV-β1 are not through VGCCs. These experiments are compelling. Thus, they show that depolarization of T cells by patch-clamp analysis did not activate VGCC currents even under optimal recording conditions, that there were no effects of TCR activation on VGCC currents, that there were no VGCC currents in T cells lacking Orai1 channels, that depolarization of T cells failed to induce Ca²⁺ influx in mouse or human T cells, and that deletion of STIM1 which can inhibit VGCCs, did not give rise to any VGCC-mediated Ca²⁺ entry. Based on these rather comprehensive analyses they conclude that there is no evidence of functional VGCCs in T cells. All these studies are extremely carefully conducted. The only slight criticism would be that it might have been better to try to eliminate both Orai1 and STIM proteins at the same time in T cells. Elimination of only Orai1 could have possibly increased the inhibitory effects of endogenous STIM1 on CaV channels, and elimination of STIM1 alone might have effects masked by the presence of Orai1 and possibly STIM2. However, this is a minor point and the overall conclusion of all the studies is compelling.

Response: Thank you for the very positive comments. To address the question whether “elimination of STIM1 alone might have effects masked by the presence of Orai1 and possibly STIM2”, we investigated whether STIM1 deletion in T cells results in a (compensatory) upregulation of ORAI1 (or other ORAI homologues) or STIM2. We did not find any upregulation of ORAI1, ORAI2, ORAI3 or STIM2 at the transcriptional level in mouse or human T cells lacking STIM1 (see below expression of CRAC channel genes from human CD4⁺ T cells isolated from a healthy donor and a patient with a null mutation that abolishes STIM1 protein expression and SOCE). Conversely, we analyzed our RNA-Seq data from CD4⁺ T cells of *Orai1^{fl/fl} Cd4Cre mice* for evidence of upregulation of STIM1 or STIM2, but that was not the case. Although unlikely in our opinion, it is possible that deletion of ORAI1 might, instead of resulting in compensatory upregulation of VGCC channels as we had postulated, free up STIM1 proteins that would otherwise bind to ORAI1 to bind instead to Cav1.2. We considered this possibility but found it to be too unlikely, especially since Cav1.2 is not expressed in human or mouse T cells and an inhibitory effect of STIM1 on other VGCCs has not been demonstrated.

The paper derives perhaps most of its significance from showing that there are no functional VGCCs in T cells. This conclusion is as important, perhaps more so, that revealing the action of CaV-β1 on T cell function. Thus, in the earlier papers, functional VGCCs have been claimed to be present in T cells and exert effects on Ca²⁺ signals and T cell function. Moreover, the earlier studies on other CaV-β subunits and their role in T cell development/survival/function have been ascribed to their role in controlling function and/or expression of VGCCs. Whereas, the negative results on VGCC involvement in the current paper could be argued as less compelling than the “positive” role of VGCCs in T cells in the earlier papers, the current paper carries with it one additional and persuasive punch. Thus, the authors show that although T cells do contain VGCC transcripts and even perhaps protein, their expression profile related to splice variations present, indicates that they are not full-length or functional proteins. Certainly the evidence for α1 pore units lacking an N-terminal region are exclusively expressed in T cells is important. Also, the authors provide good evidence that splice changes in other VGCCs render their expression of functional channels quite unlikely. So this provides a compelling, albeit provocative, message for the paper since it indicates that the rather small actions of VGCCs on Ca²⁺ changes in T cells may be spurious.

Response: We appreciate the thorough analysis of the reviewer and positive feedback. The finding of splice variants of Cav3.3 and other VGCCs in T cells was surprising and provides a compelling explanation for the fact that previous studies have reported mRNA expression of VGCCs by PCR. During the review of our manuscript, a study was published by Man et al. in *Circulation* (2021; 144:229-242), which shows that cardiomyocytes express a short transcript of the neuronal voltage-gated sodium channel Nav1.8 (Scn10a) comprising only the last 7 exons. In the case of cardiac Nav1.8, transcription of the short variant occurs from an intronic enhancer-promoter complex, whereas full-length Scn10a transcripts were undetectable in the human and mouse heart. These findings are reminiscent of the expression of Cav3.3 in human T cells and we have added a few sentences to the Discussion referring to the study by Man et al.

Whereas all the above provides quite compelling justification for the conclusion of the paper, one could argue that, given the apparent significance of the CaV-β1 subunit in clonal T cell expansion after viral infection and also in T cell apoptosis, it would have enhanced the impact of the paper if it had also included some mechanistic insights on the possible actions of CaV-β1. This is particularly true, since it is now being suggested that the actions of the other CaV-β subunits in T cells may also be independent of VGCCs. Perhaps there is some underlying role of all these subunits? Perhaps a direct comparison of the role of CaV-β1 with one or more other CaV-β subunits would be enlightening. Otherwise, it could be argued that the current paper is just adding on to the already described roles of other CaV-β subunits.

Response: We agree with the reviewer’s comment that we currently cannot explain how Cavβ1 regulates T cell function. It is noteworthy that, as we discussed in response to Reviewer 1 and in the Discussion, a role of Cavβ subunits distinct from their effects on regulating VGCC function is not a novel concept, at least not in cell types other than T cells. In the Discussion we write: “A significant body of evidence demonstrates that they interact with many other proteins and have many VGCC independent functions. For instance, Cavβ proteins interact with other ion channels including ryanodine receptors, membrane receptors, Ras-related monomeric

small GTP-binding (RGK) proteins, dynamin, actin and the scaffolding protein AHNAK1. One of the most intriguing functions of Cav β proteins is their role in controlling gene expression in the nucleus. A Cav β 4 splice variant was shown to interact with heterochromatin protein 1 γ (HP1 γ), which mediates gene silencing. Full-length Cav β 3 interacts with Pax6(S), an isoform of the transcription factor Pax6, to repress its transcriptional activity. Moreover, overexpression of Cav β 4 in HEK293 cells was shown to modulate gene expression. The Cav β 1a isoform was shown to localize to the nucleus of muscle progenitor cells (MPC) and bind to the myogenin promoter. Deletion of Cav β 1a altered MPC expansion *in vitro* and *in vivo*, and changed global gene expression. It is possible that Cav β 1 plays similar roles in T cells by controlling gene expression, which would be distinct from its regulatory role of VGCC function in excitable cells.” It is therefore quite possible that Cav β 1 plays similar roles in T cells.

We think, however, that finding out how Cacnb1 (Cav β 1) regulates T cell expansion during viral infection independent of regulating VGCCs entails an open-ended, extensive investigation that is beyond the scope of this study, which is already rather comprehensive in scope.

Overall, this reviewer believes that the paper, even without mechanistic studies, does provide an important advance in rethinking the role of VGCCs in T cell function. And, as the authors state, the conclusion that the lack of functional VGCCs in T cells provides good evidence that VGCC blockers are unlikely to have off-target effects on immune cell function, is also one of some significance.

Response: We are glad the reviewer concurs with us that our findings are important in rethinking the role of VGCCs in T cells.

One more minor point. There is a lot of repetition from the results in the discussion. Although it would take some effort to combine these, it would significantly enhance the paper’s readability.

Response: Thank you for pointing out this stylistic shortcoming of the paper. We have significantly shortened some sections of the Discussion to remove as much redundancy in the Results and Discussion sections as possible.

REVIEWERS' COMMENTS

Reviewer #1 (Remarks to the Author):

The authors have improved the manuscript and I have only positive comments followed by minor suggestions.

Comments

1) As requested, significant new data (new Supp Figures 4 and 5) includes positive controls under the same recording conditions, to demonstrate that the calcium and patch clamp assays are sufficient to detect voltage-gated calcium rises and currents in HEK cells transfected with Cav1.2 subunits, and in native PC-12 cells. This is very nice!

2) Also as requested, the authors broadened the scope of results to show that VGCC are not functionally expressed in naïve and acutely activated mouse T cells (Figure 5 new panels C and D), and are also not detected in Th2, Th17, and iTreg subsets (new Supplemental Figure 10 panels E and F). The new data represent an important enhancement of the paper. I accept the argument not to pursue this in Jurkat transformed T cells.

3) The authors tightened up the writing in the Discussion to good effect.

Minor points

1) The authors' rebuttal indicates that the Cahalan and Chandy 2009 review is cited in the Introduction on p. 5. However, the sentence remains unchanged and the citation is not in the Reference list. "A role of VGCCs in T cells, however, is not universally accepted, and biophysical evidence of VGCC currents in T cells is limited 17,19."

2) Also on p. 5 at the bottom, "We conclude that although several $\alpha 1$ and β subunits of VGCCs are expressed in T cells, they do not function as canonical VGCCs." This should be rewritten, since subunits are not expressed if the transcripts are incomplete.

3) Despite all the new data, the Abstract remains unaltered. Would the authors consider including the new data on T cell subsets in the Abstract?

Reviewer #2 (Remarks to the Author):

The authors have fully addressed comments (3)–(9) of the original critique.

One further note: Although the manuscript has ruled out the model that Cav $\alpha 1$ subunits contribute to a functional voltage-gated calcium channel in the T cells studied, the findings do not exclude the assembly of putative truncated Cav $\alpha 1$ subunits (for example, a protein comprising domains III and IV of Cav3.3 (Figure 7I)) into a homomeric or heteromeric calcium channel that is gated by a different mechanism. The possibility seems outlandish, at first glance, but then so does the low-level expression of properly spliced partial mRNA transcripts encoding such fragments. In a strict interpretation of the data, perhaps the strong message that there are no voltage-dependent calcium channels in T cells should be balanced by an acknowledgment that this alternative possible role for Cav $\alpha 1$ subunits has not been ruled out.

Reviewer #3 (Remarks to the Author):

The authors have very thoroughly addressed my comments. Also, their responses to the other reviewer's comments were extremely thorough. The paper is considerably improved as a result.

Point by point response to reviewer comments

Reviewer #1:

Minor points

1) The authors' rebuttal indicates that the Cahalan and Chandy 2009 review is cited in the Introduction on p. 5. However, the sentence remains unchanged and the citation is not in the Reference list. A role of VGCCs in T cells, however, is not universally accepted, and biophysical evidence of VGCC currents in T cells is limited 17,19.

Response: We apologize for this oversight and have added the Cahalan and Chandy 2009 review on page 5.

2) Also on p. 5 at the bottom, "We conclude that although several α_1 and β subunits of VGCCs are expressed in T cells, they do not function as canonical VGCCs." This should be rewritten, since subunits are not expressed if the transcripts are incomplete.

Response: We have rewritten that last sentence of the introduction on page 5, which now reads " We conclude that full-length transcripts of α_1 subunits of VGCCs are not expressed in T cells, providing an explanation for the absence of VGCC currents and Ca^{2+} influx upon depolarization in T cells."

3) Despite all the new data, the Abstract remains unaltered. Would the authors consider including the new data on T cell subsets in the Abstract?

Response: Most of the new data are either positive controls (excitable cells), more T helper cell subsets and naive mouse T cells. We think these data are important, but would be too granular to mention in the abstract. Especially because the editor asked us to shorten the abstract to 150 words.

Reviewer #2:

One further note: Although the manuscript has ruled out the model that Cav β_1 subunits contribute to a functional voltage-gated calcium channel in the T cells studied, the findings do not exclude the assembly of putative truncated Cav β_1 subunits (for example, a protein comprising domains III and IV of Cav3.3 (Figure 7I)) into a homomeric or heteromeric calcium channel that is gated by a different mechanism. The possibility seems outlandish, at first glance, but then so does the low-level expression of properly spliced partial mRNA transcripts encoding such fragments. In a strict interpretation of the data, perhaps the strong message that there are no voltage-dependent calcium channels in T cells should be balanced by an acknowledgment that this alternative possible role for Cav β_1 subunits has not been ruled out.

Response: We agree with the reviewer that the possibility of a functional channel, encoded by domains III and IV of Cav3.3, "seems outlandish". Such a possibility assumes that such a truncated protein is expressed and also localizes to the plasma membrane, which we consider very unlikely. This is all we are concluding from our data and we have been very careful how we interpret our data and in not overstating our findings. In fact, we do avoid writing "there are no voltage-dependent calcium channels in T cells". Instead we are deliberately more cautious or simply describe our findings:

- Abstract: "... most 5' exons of these genes are not transcribed, likely resulting in N-terminally truncated and non-functional proteins."
- Introduction (page 5): " We conclude that full-length transcripts of α_1 subunits of VGCCs are not expressed in T cells, providing an explanation for the absence of VGCC currents and Ca^{2+} influx upon depolarization in T cells."
- Results, last sentence (page 18): " Even if these proteins were stable and properly located in the plasma membrane, they would very likely not be functional Ca^{2+} channels, providing an explanation for the absence of VGCC currents and Ca^{2+} influx upon depolarization in T cells."
- Discussion (page 20): " Together, our studies fail to provide evidence for the existence of functional VGCCs in T cells."
- Discussion (page 23): " While our study does not support the existence of functional VGCCs in T cells, it suggests that Cav β_1 has alternative, VGCC-independent functions in T cells."

Nevertheless, we have added a sentence to the Discussion to mention the possibility raised by the reviewer. On page 23 we write: " While it is theoretically conceivable that a protein comprising domains III and IV of Cav3.3 is expressed and may assemble into a homomeric or heteromeric Ca²⁺ channel that is gated by a voltage-independent mechanism, we consider this possibility to be remote."